# PPM1D mutations silence NAPRT gene expression and confer NAMPT inhibitor sensitivity in glioma

Nathan R. Fons[1,2], Ranjini K. Sundaram[2], Gregory A. Breuer[1,2], Sen Peng[3], Ryan L. McLean[2], Aravind N. Kalathil[2], Mark S. Schmidt [4], Diana M. Carvalho[5], Alan Mackay[5], Chris Jones[5], Ángel M. Carcaboso [6], Javad Nazarian[7], Michael E. Berens [3], Charles Brenner [4] & Ranjit S. Bindra [1,2]

Pediatric high-grade gliomas are among the deadliest of childhood cancers due to limited knowledge of early driving events in their gliomagenesis and the lack of effective therapies available. In this study, we investigate the oncogenic role of PPM1D, a protein phosphatase often found truncated in pediatric gliomas such as DIPG, and uncover a synthetic lethal interaction between *PPM1D* mutations and nicotinamide phosphoribosyltransferase (NAMPT) inhibition. Specifically, we show that mutant PPM1D drives hypermethylation of CpG islands throughout the genome and promotes epigenetic silencing of nicotinic acid phosphoribosyltransferase (NAPRT), a key gene involved in NAD biosynthesis. Notably, *PPM1D* mutant cells are shown to be sensitive to NAMPT inhibitors in vitro and in vivo, within both engineered isogenic astrocytes and primary patient-derived model systems, suggesting the possible application of NAMPT inhibitors for the treatment of pediatric gliomas. Overall, our results reveal a promising approach for the targeting of *PPM1D* mutant tumors, and define a critical link between oncogenic driver mutations and NAD metabolism, which can be exploited for tumor-specific cell killing.

[1] Department of Pathology, Yale University, New Haven, CT 06520, USA. [2] Department of Therapeutic Radiology, Yale University, New Haven, CT 06520, USA. [3] The Translational Genomics Research Institute (TGen), Phoenix, AZ 85004, USA. [4] Department of Biochemistry, University of Iowa, Iowa City, IA 52242, USA. [5] Divisions of Molecular Pathology and Cancer Therapeutics, Institute of Cancer Research, London, UK. [6] Institut de Recerca Sant Joan de Deu, Barcelona 08950, Spain. [7] Children's National Health System, Washington, DC 20010, USA. Correspondence and requests for materials should be addressed to M.E.B. (email: mberens@tgen.org) or to C.B. (email: charles-brenner@uiowa.edu) or to R.S.B. (email: ranjit.bindra@yale.edu)

The Protein Phosphatase $Mg^{2+}/Mn^{2+}$ Dependent 1D (PPM1D) gene, also known as Wip1, encodes a serine/threonine phosphatase which dephosphorylates numerous proteins primarily involved in the DNA damage response (DDR) and cellular checkpoint pathways[1]. Since its discovery over 20 years ago, PPM1D has become a well-established oncogene, found amplified or over-expressed in a diverse range of cancers, including breast, ovarian, gastrointestinal, and brain cancers[2–7]. Truncation mutations in the C-terminus of PPM1D were subsequently identified in a subset of cancers, most notably in pediatric gliomas, including diffuse intrinsic pontine glioma (DIPG)[8–10]. These mutations markedly enhance the stability of PPM1D, increasing its total overall phosphatase activity[11]. Despite characterization of the cellular function of this important enzyme, there remains much to be understood about the role of PPM1D in tumorigenesis. To compound this, there are no isogenic glial cell lines that contain PPM1D-truncating mutations, limiting the ability to study the specific consequences of these genomic events in the formation of gliomas. Finally, while a number of PPM1D inhibitors have been developed as promising experimental tools[12], their success in vitro has not translated into the clinic, exposing an important and unmet clinical need.

Here we describe the creation and validation of PPM1D-truncated isogenic astrocyte cell lines for use in studying the role of these mutations in gliomagenesis. Through a targeted synthetic lethal drug screen, we demonstrate that PPM1D mutant astrocytes and patient-derived PPM1D mutant DIPG lines are particularly sensitive to treatment with nicotinamide phosphoribosyltransferase (NAMPT) inhibitors. Finally, we show that this mutant PPM1D-induced NAMPT inhibitor sensitivity is driven by hypermethylation of CpG islands throughout the genome, and in particular, the epigenetic silencing of nicotinic acid phosphoribosyltransferase (NAPRT), a key gene involved in nicotinamide adenine dinucleotide (NAD) biosynthesis. These findings provide important insights into the biological effects of truncating PPM1D mutations, and uncover unique vulnerabilities associated with enhanced PPM1D activity which can be exploited for the therapeutic intervention of mutant pediatric brain tumors.

## Results

### PPM1D mutant astrocytes are sensitive to NAMPT inhibitors.
To develop PPM1D mutant models for subsequent biological investigations, we used CRISPR/Cas9 genomic editing to create isogenic immortalized human astrocytes harboring endogenous PPM1D truncation mutations (PPM1D$^{trncs.}$). The heterozygous, truncating mutations were introduced into exon 6 of the PPM1D locus, at C-terminal locations similar to those found in DIPGs (Fig. 1a). We then isolated single cell PPM1D$^{trnc.}$ clones and confirmed the presence of frameshifting mutations that encode truncated PPM1D proteins (Supplementary Fig. 1a). As expected, truncated PPM1D was highly expressed in mutant cells (Fig. 1b) and maintained a substantially longer half-life compared to the wild type (WT), full-length form of the protein (Fig. 1c, d). The increased PPM1D protein stability correlated with enhanced phosphatase activity as seen by the active dephosphorylation of key PPM1D targets, γH2AX and pCHK2 (T68), measured by western blot (Supplementary Fig. 1b) and γH2AX foci formation assays (Fig. 1e; Supplementary Fig. 1c), after exposure to ionizing radiation (IR). Importantly, these differences were abolished by treatment with GSK2830371, a known inhibitor of PPM1D[12] (Fig. 1f).

Given the role of PPM1D in DDR pathways, we performed a small molecule synthetic lethal screen with a panel of inhibitors against key DNA repair and metabolic proteins, using methodology described previously by our group[13]. This screen identified a

synthetic lethal interaction between PPM1D mutations and the NAMPT inhibitor, FK866[14] (Fig. 1g; Supplementary Table 1). This unexpected NAMPT inhibitor sensitivity was confirmed in three different PPM1D$^{trnc.}$ cell lines (Fig. 1h), as well as by three structurally distinct NAMPT inhibitors: STF31, GPP78, and STF118804 (Fig. 1i; Supplementary Fig. 1d), corroborating our initial finding and establishing that this effect is a result of on-target inhibition of NAMPT activity. Furthermore, stable over-expression of either WT or mutant PPM1D in the parental astrocyte cell line (PPM1D OE$^{FL}$ or OE$^{trnc.}$, respectively), was sufficient to confer FK866 synthetic lethality, confirming that this interaction is driven specifically by an increased total activity of PPM1D, and not a neomorphic role of the mutant protein (Fig. 1j; Supplementary Fig. 1e–g). Additionally, expression of a phosphatase dead mutant (PPM1D D314A), did not result in FK866 sensitivity in our astrocyte models, further verifying the dependence on increased PPM1D activity for the induction of this synthetic lethality.

### Reduced NAD levels in PPM1D$^{trncs.}$ drive NAMPTi sensitivity.
Next, we sought to investigate the molecular basis for mutant PPM1D-induced NAMPT inhibitor (NAMPTi) synthetic lethality. NAMPT catalyzes the rate-limiting step in the salvage of nicotinamide (NAM) to form NAD (Fig. 2a)[14,15]. Thus, we wished to quantify the NAD metabolome[16], within our WT and PPM1D$^{trnc.}$ astrocyte models to better understand potential variations in this important metabolic pathway. We found that PPM1D mutations induce a substantial depression of many NAD-related metabolites, including a significant reduction in NAD and NADP levels (Fig. 2b, c; Supplementary Fig. 2a). As maintenance of these two cofactors is important for cellular bioenergetics and proliferation[15,17], we examined the effect of NAMPT inhibition on the quantities of both NAD and NADP, as well as on cell viability. While cellular pools of both NAD and NADP dropped markedly in FK866-treated WT astrocytes, the decline was significantly greater in the PPM1D$^{trnc.}$ cells (Fig. 2d; Supplementary Fig. 2b), indicating an enhanced dependence on NAMPT activity in the setting of mutant PPM1D. We then tested whether nicotinamide riboside (NR) could bypass NAMPT inhibition and thus, rescue the levels of NAD in PPM1D$^{trnc.}$ astrocytes[18]. Indeed, NR treatment sufficiently increased basal NAD levels (Supplementary Fig. 2c, d), and completely mitigated the cytotoxic effects of FK866 in PPM1D$^{trnc.}$ cells (Fig. 2e; Supplementary Fig. 2e, f). Similar results were found upon exogenous treatment of NAM, which strongly antagonized FK866 cytotoxicity in PPM1D$^{trnc.}$ cells (Supplementary Fig. 2g–i). Interestingly, exogenous treatment with NA did not prevent FK866-induced cell death, indicating a potential metabolic defect in the Preiss Handler salvage pathway (Supplementary Fig. 2j–l). Taken together, these data suggest that mutant PPM1D induces a depression of the NAD metabolome and especially NAD levels, which can be further potentiated by NAMPT inhibition, resulting in the selective killing of PPM1D mutant cells.

### PPM1D mutant DIPG models silence NAPRT gene expression.
To understand the underlying cause of NAD depletion in PPM1D$^{trnc.}$ cells, we performed a focused synthetic lethal siRNA screen in our isogenic astrocytes, targeting key enzymes involved in NAD synthesis and consumption pathways. Using FK866 sensitivity as an endpoint, the goal was to identify genes whose loss phenocopies the synthetic lethal interaction previously identified between mutant PPM1D and NAMPT inhibition. From this screen, we found that siRNA-mediated knockdown of NAPRT induced profound sensitivity of the parental astrocyte cell line to FK866 treatment (Fig. 2f; Supplementary Fig. 3a).

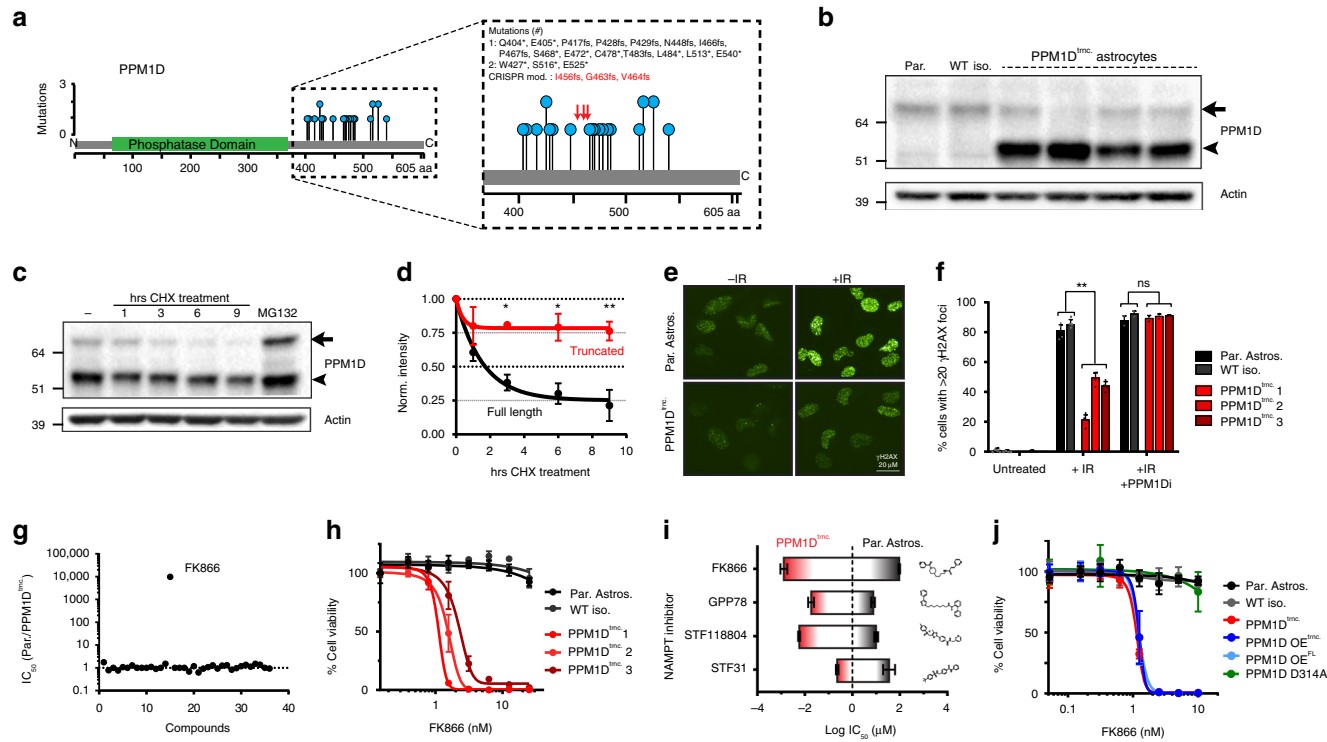

**Fig. 1** *PPM1D* mutant immortalized human astrocytes are sensitive to NAMPT inhibitors. **a** Previously identified (refs. [8-10]) *PPM1D* truncation mutations in pediatric HGGs (blue circles). CRISPR-modified mutations in human astrocytes shown in red arrows. **b** Immunoblot of PPM1D full-length (full arrow) and truncated (arrowhead) protein expression across parental astrocytes (Par.), an isolated wild type astrocyte clone (WT iso.), and four different isolated CRISPR-modified, PPM1D-truncated (PPM1D$^{trnc.}$) astrocytes. **c** Immunoblot of PPM1D expression post cycloheximide (CHX) and MG132 treatment. **d** Quantification of the experiment in (**c**), ($n = 3$ biologically independent experiments, *$p < 0.05$, **$p < 0.01$ by Student's $T$ test). **e** Representative images of cellular γH2AX foci, $+/-$ treatment with 10 Gy ionizing radiation (IR). **f** Quantification of γH2AX foci in untreated, IR-treated, and concurrent IR plus 50 nM PPM1D inhibitor GSK2830371 treatment (PPM1Di); ($n = 4$ biologically independent samples, **$p < 0.001$ by Student's $T$ test). **g** Calculated IC$_{50}$ ratios (Parental/PPM1D$^{trnc.}$) for a library of tested small molecule inhibitors. **h** Viability assessment of wild type (Par. Astros. and WT iso.) and three PPM1D$^{trnc.}$ cell lines, 72hrs post FK866 treatment ($n = 3$ biologically independent samples). **i** Calculated IC$_{50}$ values of parental (black highlight) and PPM1D$^{trnc.}$ (red highlight) astrocytes for different NAMPT inhibitors; length of bar represents selectivity window of the given drug for *PPM1D* mutant cells ($n = 2$ biologically independent experiments). **j** Viability analysis of cell lines in response to 72 h of FK866 treatment ($n = 3$ biologically independent samples). All error bars represent standard deviation of the mean

Additional NAPRT siRNAs were used to confirm these findings and further revealed a strong correlation between the degree of NAPRT knockdown and FK866 sensitivity (Supplementary Fig. 3b, c). NAPRT plays a complementary role to NAMPT in the production of NAD, and previous studies have inversely correlated NAPRT expression with NAMPT inhibitor sensitivity[19-22]. Surprisingly, we found that NAPRT protein expression was undetectable in our PPM1D$^{trnc.}$ and PPM1D overexpressing (OE$^{FL}$ and OE$^{trnc.}$) cell lines (Fig. 2g). To determine if this critical deficiency resulted in NAMPT inhibitor sensitivity, we reintroduced NAPRT into PPM1D$^{trnc.}$ cells. Stable, ectopic expression of NAPRT completely rescued the cytotoxicity caused by NAMPT inhibition and mirrored the resistance found commonly in WT cells (Fig. 2h; Supplementary Fig. 3d, e). Collectively, these findings suggest that mutant PPM1D drives a loss of NAPRT expression, which ultimately induces profound NAMPT inhibitor sensitivity.

To complement our work in immortalized, normal human astrocytes, we then tested whether our findings could be recapitulated in more clinically relevant tumor models. To this end, we examined NAPRT expression in a collection of previously described, patient-derived DIPG spheroid cultures[9,23]. One of these DIPG lines, SU-DIPG-XXXV, contained a S432fs mutation in *PPM1D* (Supplementary Fig. 4a, b), and prominently expressed a hyperstable, truncated form of the protein (Fig. 2i). Similar to

the PPM1D$^{trnc.}$ astrocytes, we found that SU-DIPG-XXXV also completely lacked NAPRT gene expression. This deficiency was unique in the DIPG cell panel as the remaining WT lines maintained high levels of NAPRT expression. Consistent with our findings in immortalized astrocytes, SU-DIPG-XXXV was also extremely sensitive to FK866 treatment (Fig. 2j, k) with cytotoxic doses in the low, single-digit nanomolar range. In contrast, the three WT DIPG lines were resistant to FK866 treatment, highlighting the dependence of NAMPT inhibitor sensitivity on *PPM1D* mutation status. Notably, culturing these DIPG cell lines in growth media devoid of nicotinic acid (NA) induced a strong sensitivity to FK866 in all SU-DIPG spheroid cultures tested (Supplementary Fig. 4c), confirming the importance of alternative NAD biosynthesis pathways such as NA salvage, in mediating NAMPT inhibitor synthetic lethality in gliomas.

**Epigenetic events silence NAPRT expression in PPM1D mutant models.** Next we sought to identify the mechanism by which mutant PPM1D suppresses NAPRT expression. While *NAPRT* mRNA was highly expressed in WT DIPG lines (SU-DIPG-IV, XIII, and XVII), *NAPRT* transcript levels were found to be significantly depressed in all PPM1D mutant astrocyte and DIPG models tested (PPM1D$^{trnc.}$, PPM1D$^{OE}$, and SU-DIPG-XXXV) (Fig. 3a), indicating the presence of a conserved

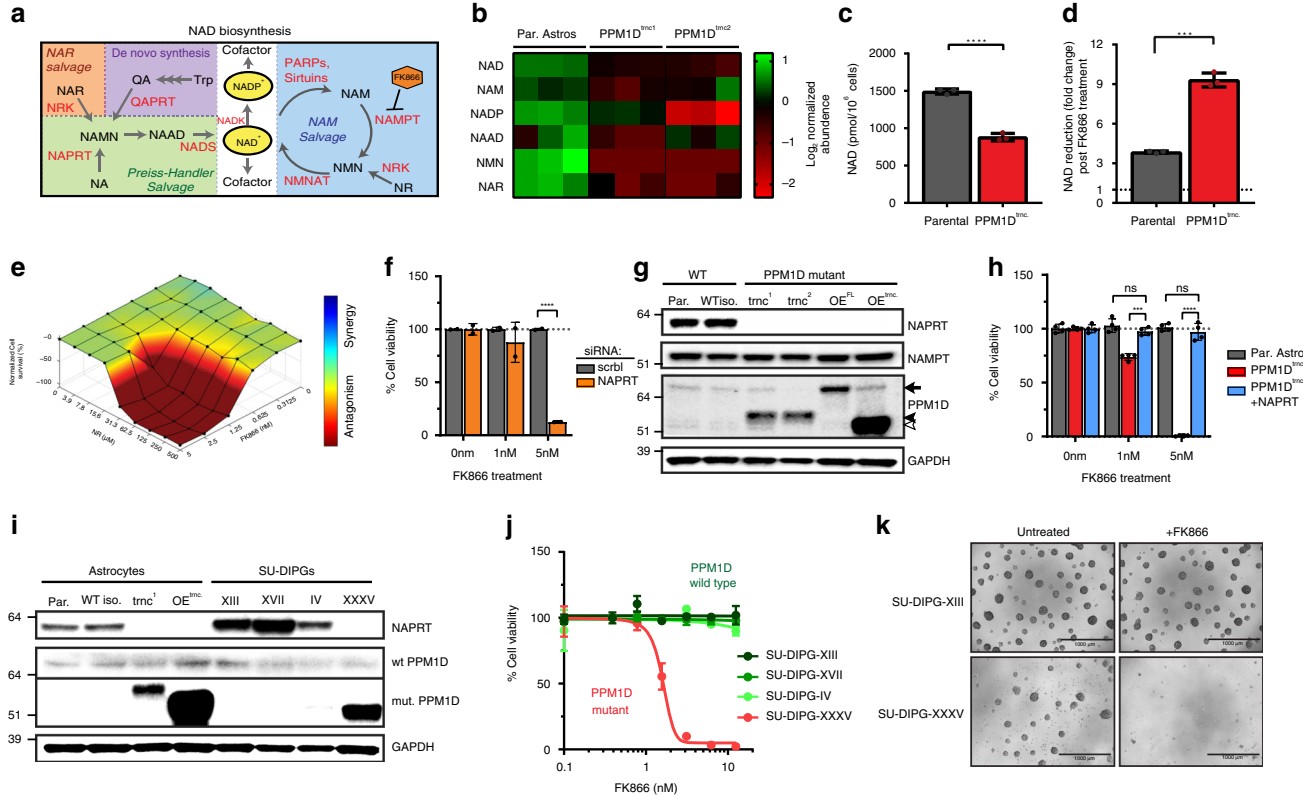

**Fig. 2** Mutant PPM1D-induced NAPRT deficiency drives sensitivity to NAMPT inhibition. **a** Graphic model of enzymes (red) and metabolites (black) involved in NAD biosynthesis. NA nicotinic acid, NAAD nicotinic acid adenine dinucleotide, NAD nicotinamide adenine dinucleotide, NADP nicotinamide adenine dinucleotide phosphate, NAM nicotinamide, NAMN nicotinic acid mononucleotide, NAR nicotinic acid riboside, NMN nicotinamide mononucleotide, NR nicotinamide riboside, QA quinolinic acid, Trp tryptophan. **b** Heatmap of NAD-related metabolites in parental and two different PPM1D[trnc.] astrocyte cell lines. **c** NAD quantification in wild type and PPM1D[trnc.] astrocytes ($n = 3$ biological independent samples, ****$p < 0.0001$ by Student's $T$ test). **d** Relative fold change in NAD levels post 10 nM FK866 treatment ($n = 3$ biological independent samples, ***$p < 0.001$ by Student's $T$ test). **e** Bliss 3D surface plot modeling the antagonistic effects of NR on FK866 treatment in PPM1D[trnc.] astrocytes. **f** Cell viability analysis of parental astrocytes treated with either scrambled control (scrbl) or NAPRT siRNAs, followed by treatment with FK866 ($n = 2$ biological independent samples, ****$p < 0.0001$ by Student's $T$ test). **g** Immunoblot of isogenic astrocytes., and astrocytes stably-overexpressing WT and mutant PPM1D (OE[FL] and OE[trnc.], respectively). Full length (full arrow), CRISPR-modified (black arrowhead), and ectopic mutant (white arrowhead) sizes of PPM1D displayed. **h** Viability assessment of isogenic astrocytes and stable NAPRT-expressing PPM1D[trnc.] astrocytes (PPM1D[trnc.] + NAPRT), after FK866 treatment ($n = 4$ biological independent samples, ***$p < 0.001$, ****$p < 0.0001$ by Student's $T$ test). **i** Immunoblot of previously described wild type and *PPM1D* mutant astrocytes, and patient-derived, SU-DIPG cell lines. **j** Viability assessment of SU-DIPG cell lines post 120 h treatment with FK866 ($n = 3$ biological independent samples). **k** Representative images from spheroid cultures in (**j**), untreated or treated with 10 nM FK866. All error bars represent standard deviation of the mean

transcriptional repression of the *NAPRT* gene. As transcriptional silencing is often controlled by epigenetic factors, we next examined the occupancy of different histone marks at the *NAPRT* promoter in WT and *PPM1D* mutant astrocytes. Using chromatin immunoprecipitation (ChIP), we found that transcriptional repression of *NAPRT* in *PPM1D* mutant cells correlated with a substantial loss in key activating chromatin marks, H3K4me3 and H3K27ac (Fig. 3b). It has previously been shown that a loss of occupancy of H3K4me3 and H3K27ac is associated with an increase in site-specific DNA methylation[24–26]. Additionally, the *NAPRT* promoter resides within a CpG island that is prone to de novo DNA methylation[21,27]. Thus, we considered the possibility that mutant PPM1D induces silencing of the *NAPRT* gene by regulating DNA methylation at its promoter. To test this hypothesis, we immunoprecipitated and quantified methylated and hydroxymethylated cytosine bases from within the *NAPRT* promoter, using Me-DIP and hMe-DIP assays respectively. From this work we detected a prominent increase in DNA methylation, but not hydroxymethylation, at the *NAPRT* promoter in PPM1D[trnc.] astrocytes (Fig. 3c). This finding was further confirmed with bisulfite conversion and sequencing of our astrocyte

and DIPG models, which revealed extensive *NAPRT* promoter hypermethylation in all *PPM1D* mutant cell lines (Fig. 3d). To ascertain if this effect was specifically limited to DIPG and astrocyte models, we validated our results in the osteosarcoma cell line, U2OS (R458fs), as well as the breast cancer cell line MCF7 (*PPM1D* amplification), both which contain endogenous *PPM1D* alterations (Supplementary Fig. 5a, b). Similar to the PPM1D[trnc.] astrocytes, we found substantial gene silencing of *NAPRT* transcription in U2OS and MCF7 cells, which corresponded with extensive hypermethylation of the *NAPRT* promoter (Supplementary Fig. 5c, d). Further, both cell lines displayed a strong sensitivity to FK866 treatment, which was comparable to PPM1D[trnc.] astrocytes and the other described *PPM1D* mutant DIPG models (Supplementary Fig. 5e).

**PPM1D mutations promote global CpG island hypermethylation**. Next, we investigated whether mutant PPM1D-induced *NAPRT* gene silencing is a focal event or part of a more global phenomenon. Whole genome methylation profiling was performed on our entire panel of WT and *PPM1D* mutant cell lines,

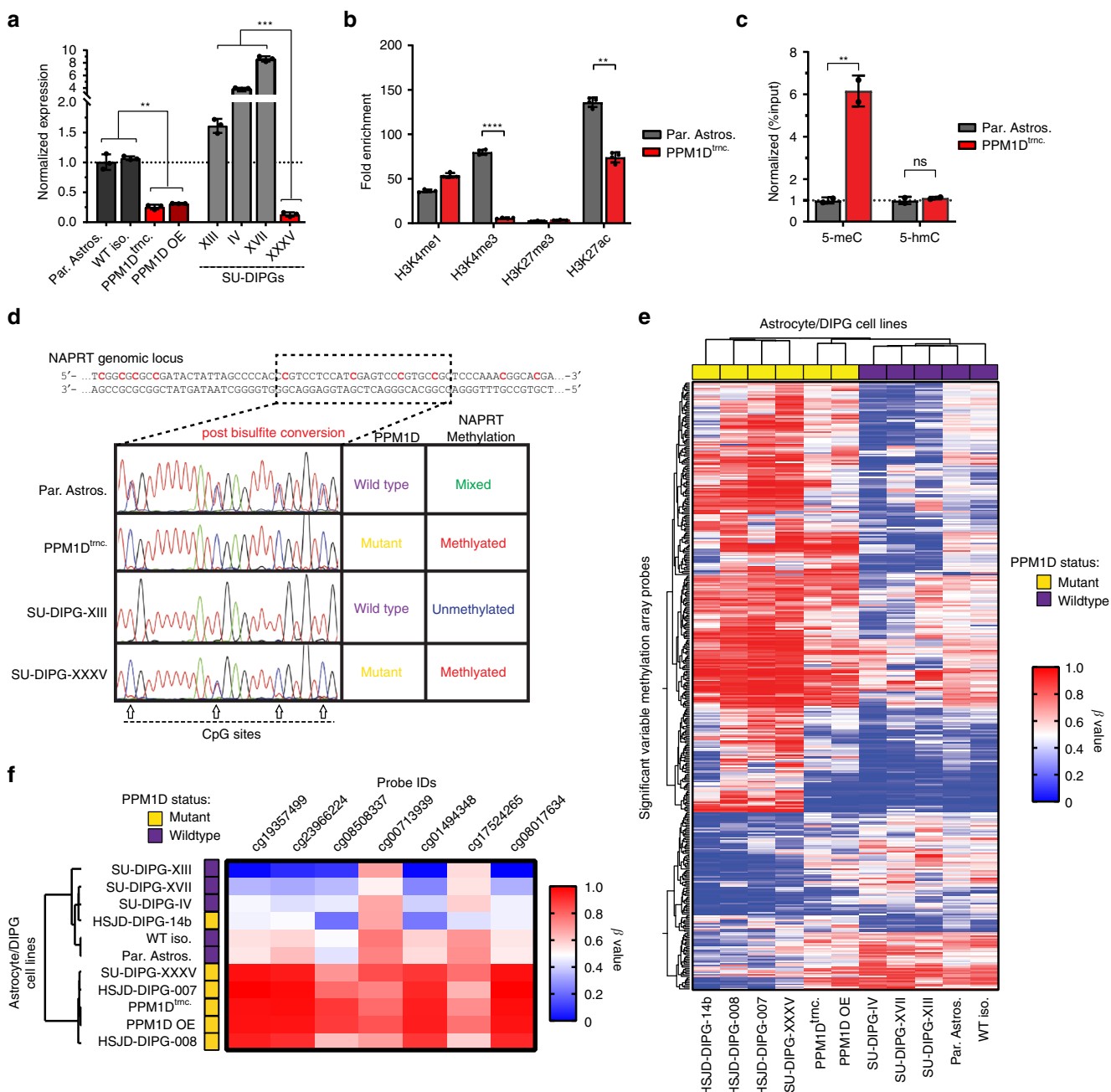

**Fig. 3** Epigenetic events silence *NAPRT* expression in *PPM1D* mutant glioma models. **a** Quantification of *NAPRT* transcript levels via qPCR, in wild type (gray) and mutant PPM1D-expressing (red) astrocytes and DIPG cell lines ($n = 3$ biological independent samples, **$p < 0.01$, ***$p < 0.001$ by Student's $T$ test). **b** Chromatin Immunoprecipitation (ChIP) of common histone 3 modifications at the *NAPRT* promoter; quantified as fold enrichment over IgG control ($n = 4$ biological independent samples, **$p < 0.01$, ****$p < 0.0001$ by Student's $T$ test). **c** Quantification of methylated DNA (5-meC), and hydroxymethylated DNA (5-hmC), immunoprecipitated from the *NAPRT* promoter ($n = 2$ biological independent samples, **$p < 0.01$ by Student's $T$ test). **d** Sequencing chromatograms of the *NAPRT* promoter within astrocytes and SU-DIPG cell lines after bisulfite conversion; arrows indicate potential CpG methylation sites. **e** Heatmap and clustering analysis of the 390 most significant variable Infinium Methylation EPIC array probes, across different astrocyte and DIPG models. **f** Heatmap and hierarchical clustering analysis of methylation array probes located within *NAPRT* CpG island promoter region. All error bars represent 95% confidence intervals about the mean

as well as on three additional *PPM1D* mutant DIPG lines: HSJD-DIPG-007, HSJD-DIPG-008, and HSJD-DIPG-14b; all of which maintain reduced expression of NAPRT[28] and/or sensitivity to FK866 treatment (Supplemental Fig. 6a–d). Methylation results from the Illumina Human EPIC Bead Array (850 k) revealed a substantial increase in CpG island hypermethylation across all *PPM1D* mutant cell lines tested. Of the 390 most significant variable probes (SVPs), 287 (74%) were hypermethylated in

*PPM1D* mutant lines (PPM1D^trnc., PPM1D^OE, SU-DIPG-XXXV, HSJD-DIPG-007, HSJD-DIPG-008, and HSJD-DIPG-14b), compared to only 103 (26%) hypermethylated in WT cell lines (Fig. 3e). In addition, individual probes within the *NAPRT* locus were subsequently identified and analyzed from this data set. All seven sites residing within the CpG island promoter region of *NAPRT* were heavily methylated in *PPM1D* mutant astrocytes and DIPG cultures, and bivariate correlational analysis clustered 5

of 6 mutant cells separately from tested WT lines (Fig. 3f). Interestingly, despite a lower overall degree of methylation within the *NAPRT* promoter in HSJD-DIPG-14b, this line exhibted hypermethylation across the SVPs described previously, and clustered similarly to the other *PPM1D* mutant lines upon whole genome methylation analysis. Of note, all DIPG lines tested harbored endogenous histone 3 K27M mutations (Supplemental Fig. 6a, e), which often co-occur with PPM1D truncating mutations in tumor samples[10,29]. Despite previous reports linking H3.1 or H3.3 K27M mutations to global DNA hypomethylation, our results suggest that truncation alterations in PPM1D may in fact overcome this effect, and instead drive the hypermethylation of genomic CpG islands.

IDH1 R132H mutant gliomas famously exhibit a glioma-associated CpG island methylator phenotype (or G-CIMP), which arises from the competitive inhibition of DNA-demethylating TET proteins by the oncometabolite 2-HG[30]. To understand if the hypermethylation events observed in our *PPM1D* mutant DIPG models paralleled those found in *IDH1* mutant cell lines, we analyzed the top 2% of significantly variable CpG island methylation array probes for comparison to a previously published IDH1 mutant data set from Turcan et al.[30] (Supplementary Fig 7a, b.) While we identified a similar percentage of hypermethylated probes in the *PPM1D*- and *IDH1* mutant cell lines compared to their parental astrocyte controls (79.4% and 63.9%, for PPM1D mutant- and IDH1 mutant astrocytes, respectively) we found surprisingly little over-lap between the two engineered mutant lines (Supplementary Fig. 7c). Further, examination of global 5-hydroxymethylcytosine (5-hmC), a product of TET enzymatic activity, found no significant difference in 5-hmC levels between WT and *PPM1D* mutant astrocytes, indicating a distinct mechanism may be driving the development of genomic hypermethylation in these mutant cell lines (Supplemental Fig. 7d). Lastly, treatment of PPM1D$^{trnc.}$ cells with the DNA demethylating agents decitabine (DCT) and azacytidine (azaC) failed to reverse the gene silencing of NAPRT in these cells, further differing our results from previous studies in *IDH1* mutant cell lines[31] (Supplemental Fig. 7e). Overall, these findings demonstrate that *PPM1D* mutations drive a unique pattern of global DNA methylation, distinct from that found in *IDH1* mutant gliomas, which is associated with CpG island hypermethylation and NAPRT gene silencing.

**NAMPTis are efficacious in vivo against PPM1D$^{mut}$ xenografts**. Next, we tested whether mutant PPM1D-induced NAMPT inhibitor sensitivity could be recapitulated in vivo. We subcutaneously injected both parental and PPM1D$^{trnc.}$ cells into NOD *scid* gamma (NSG) mice and monitored tumor growth using bioluminescence imaging (BLI). While parental astrocytes failed to form tumors after 6 months, flank injection of PPM1D$^{trnc.}$ astrocytes resulted in tumor formation within 30 days. Remarkably, treatment of these mice with FK866 induced a rapid reduction in tumor burden (fold change = 4.93, $p = 0.0003$ by Mann–Whitney $U$ test) after 3 weeks (Supplementary Fig. 8a, b). These data correlated with substantially lower (fold change = 3.1, $p < 0.0001$ by Mann–Whitney U test) final tumor mass after treatment with FK866 versus vehicle alone (Supplementary Fig. 8c). As the size and growth rate of PPM1D$^{trnc.}$ xenografts limit the use of alternative measurement techniques, we created a serially-transplanted, *PPM1D* mutant astrocyte xenograft model. These *PPM1D* mutant xenografts form measurable tumors within 12 days of flank injection (Supplementary Fig. 8d) and grow rapidly, allowing direct tumor volumes to be assessed. Treatment of these mice with FK866

greatly reduced the overall tumor size (fold change = 17.1, $p < 0.0002$ by Mann–Whitney $U$ test), as measured by both calipers and BLI, (Fig. 4a; Supplementary Fig. 8e), and significantly delayed tumor growth ($p < 0.0001$ by Log rank (Mantel-Cox) test) compared to a vehicle control (Fig. 4b). Similar results were obtained in U2OS cell line xenografts, which again displayed hypersensitivity to FK866 treatment (fold change = 5.86, $p < 0.0001$ by Mann–Whitney $U$ test) (Supplementary Fig. 9a). Importantly, as NAMPT inhibitors have been associated with dose-related toxicities, the health and body mass of all mice on study were tracked throughout the dosing schedule, during which time we detected no significant differences in body mass between the treatment groups (Supplementary Fig. 9b). Overall, our data strongly support the synthetic lethality seen with FK866 in vitro, and demonstrate the potential efficacy of NAMPT inhibitors for treatment of *PPM1D* mutant tumors.

Finally, using gene expression data from within a cohort of DIPG biopsy specimens[32], we identified a strong inverse correlation between *PPM1D* and *NAPRT* mRNA levels (Supplementary Fig. 9c), as well as a trend of decreased *NAPRT* expression in known *PPM1D* mutant tumor samples (Fig. 4c; Supplementary Fig. 9d). In parallel, we analyzed publicly available patient-derived cancer gene expression data from cBioPortal[33,34] across tumor subtypes in which *PPM1D* is often found amplified, including brain, breast, and ovary[3–7]. From this, we identified a trend of statistically significant differences in *NAPRT* expression between *PPM1D* low and high expressing tumors (Supplementary Fig. 9e), providing additional validation across a diverse set of malignancies that associates expression of this oncogene with a potentially actionable and druggable target.

## Discussion

Altogether, our results establish a previously unknown role for *PPM1D* mutations as drivers of global DNA methylation, leading to *NAPRT* gene silencing. NAPRT catalyzes the first step in the Preiss-Handler NA salvage pathway to produce NAD. Thus, mutant PPM1D-induced silencing of *NAPRT* leads to a depression of the NAD metabolome. Loss of NAPRT necessitates a complete reliance of *PPM1D* mutant cells on other NAD-generating pathways for survival, principally the NAM-salvage pathway mediated by NAMPT. As a result, *PPM1D* mutant cells can be selectively targeted and killed with NAMPT inhibitors (Fig. 4d). Additionally, NAMPT inhibitor synthetic lethality was observed in an assorted panel of cells expressing high levels of both truncated or full-length PPM1D. This finding suggests broad clinical applicability, since *PPM1D* is amplified or over-expressed in a diverse range of cancers[2–7].

A previous study unexpectedly identified the occurrence of *PPM1D* mutations in adult and pediatric brainstem gliomas, yet failed to detect the presence of mutant PPM1D-induced hypermethylation events[10]. This result, however, is not unexpected as the focus of this previous work was to characterize the genetic landscape of brainstem and thalamic gliomas, and thus, utilized an assorted panel of tumors possessing a variety of potentially confounding mutations, including a significant number of *IDH1* mutant samples. As only five *PPM1D* mutant tumors were present in this diverse cohort (5 of 45 total samples; 11%), the whole genome methylation profiling and analysis performed in this study was likely not sufficient to specifically identify differences in DNA methylation patterns in these *PPM1D* mutant gliomas. Therefore, our work which uses a variety of engineered-isogenic astrocytes, in addition to well-characterized, patient-derived DIPG neurosphere models, provides a unique and thorough investigation into the DNA methylation alterations induced by truncation mutations in *PPM1D*.

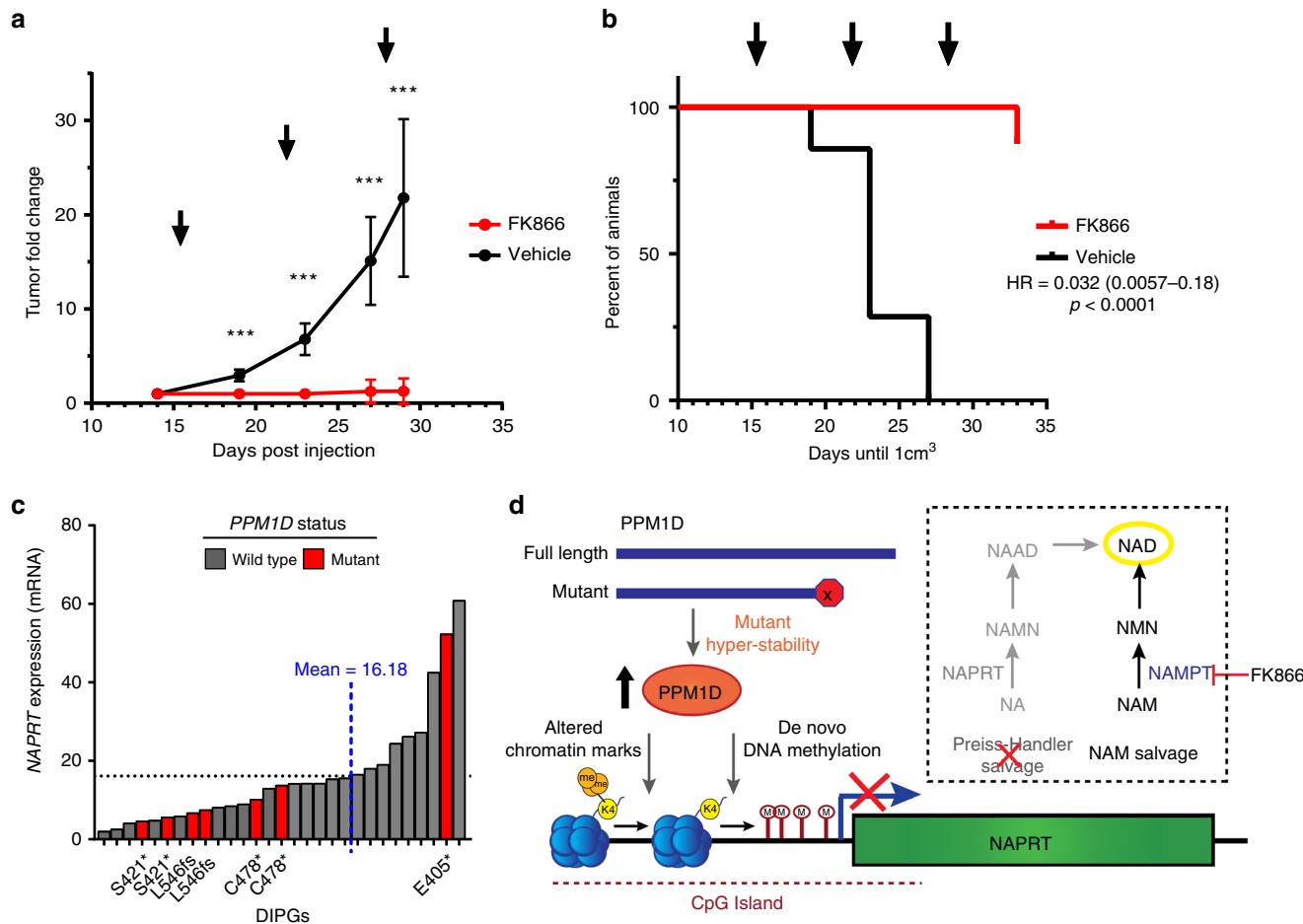

**Fig. 4** NAMPT inhibitors are effective in vivo agents against *PPM1D* mutant xenografts. **a** Fold change in tumor growth for serially-transplanted PPM1D[trnc.] xenografts in NSG mice treated with vehicle or 20 mg/kg FK866 BID for 3 cycles of: 4 days on, followed by 3 days off (*n* = 7 animals, ***p < 0.001 by Mann–Whitney *U* test, error bars represent standard deviation of the mean). Arrows indicate initiation of treatment cycle. **b** Kaplan–Meier plot of xenograft tumor growth from (**a**), with arrows indicating initiation of treatment cycle (*p* < 0.0001 by Log rank (Mantel–Cox) test). **c** *NAPRT* expression levels for PNOC003 DIPG cohort[31] samples. **d** Model depicting the mechanism of mutant PPM1D-induced dependence on NAMPT for NAD production, and synthetic lethality with NAMPT inhibitors, such as FK866

NAMPT inhibitors have been tested in clinical trials, although the lack of a prognostic biomarker, as well as dose-limiting hematologic toxicities, have stymied their further advancement into the clinic[35]. Our study reveals a clinically-relevant bio-marker, *PPM1D* mutations, which can be used for molecularly-informed personalized treatment of patients using NAMPT-inhibitor based therapeutic strategies. Furthermore, previous studies suggest that numerous DNA damaging agents, such as temozolomide[36] and radiation therapy[37], also deplete cellular levels of NAD. As these agents are commonly used to treat tumors that harbor *PPM1D* mutations (e.g., DIPG), they could be combined with NAMPT inhibitors to further enhance tumor-selective cytotoxicity. Recent reports suggest that co-administration of NA can mitigate NAMPT inhibitor-associated hematologic toxicity via the production of NAD through the NA salvage pathway[38]. Based on our observations that mutant PPM1D blocks this pathway via tumor-specific *NAPRT* silencing, NA supplementation may be an effective approach to further enhance the therapeutic index associated with NAMPT inhibition. Finally, our results reveal a unique pattern of CpG island hypermethylation events, specifically in DIPGs. This finding is reminiscent yet biologically distinct from that associated with *IDH1/2* mutations in adult gliomas[30]. Overall, our work demonstrates a completely independent route by which tumor-

associated mutations can drive global DNA hypermethylation events, and sheds additional light on the molecular consequences of aberrant methylation in glioma biology.

## Methods

**Cell culture materials and techniques**. hTert/E6/E7 immortalized human astrocytes were acquired from the lab of Dr. Timothy Chan, and have been previously characterized[30,39]. Unless noted otherwise, astrocytes were grown in DMEM, high glucose (ThermoFisher Scientific/Gibco) plus 10% FBS (Gibco) as adherent monolayers. U2OS cells were purchased from ATCC, and were grown in DMEM, high glucose plus 10% FBS. MCF7 cells were a gift from Dr. Peter Glazer, and grown in RPMI1640 (ThermoFisher Scientific/Gibco) with the addition of 10% FBS. SU-DIPG spheroid cell lines were acquired from the lab of Dr. Michelle Monje. The HSJD-DIPG-007 and HSJD-DIPG-008 neurosphere lines were acquired from Chris Jones and Angel M. Carcaboso. HSJD-DIPG-007, HSJD-DIPG-008, and SU-DIPGs lines were all cultured in a Tumor Stem Media Base (DMEM/F12 and Neurobasal media) with the addition of growth factors: B27 supplement (Gibco/ThermoFisher), human EGF (Sigma), human FGF (Sigma), human PDGF (Sigma), heparin (Stemcell Technologies), and with or without the addition of nicotinic acid (Sigma), as indicated.

**CRISPR/Cas9 genomic editing and plasmids**. CRISPR/Cas9 genomic editing was performed in astrocytes using expression of both Cas9 (Addgene #43861) and a modified guide RNA (gRNA) construct (Addgene #43860), both gifts from Dr. Keith Joung[40]. *PPM1D* gRNA sequences are available in Supplementary Table 2 and were synthesized, annealed, and ligated into the gRNA plasmid. Both constructs were then co-transfected into astrocytes through nucleofection (Lonza), and

the cells were incubated for 72 h prior to harvest and isolation. Isolated clones were generated through a single cell dilution approach, and were grown up from individual wells of a 96-well plate. Clone screening for mutant PPM1D sequences and expression was performed using high resolution melt analysis screening methods[41] and by western blotting as described below.

**Creation and integration of expression constructs.** An hWIP1 wild type plasmid was a gift from Lawrence Donehower (Addgene # 28105)[42]. PPM1D was then subcloned from hWIP1 into a modified-phCMV1 expression construct (a gift from Dr. Ryan Jensen) creating PPM1D OE$^{FL}$. This construct was modified using site-directed mutagenesis, with the primers listed in Supplementary Table 2, to introduce an R458fs mutation, creating PPM1D OE$^{trnc}$. All constructs were amplified in E.coli and purified using a MidiPrep kit (Qiagen), for nucleofection into cell lines as described above. Stable cell lines were selected with G418 (Gibco/ThermoFisher), and further isolated from single cell cultures. hWIP1 D314A phosphatase dead expression construct (from Lawrence Donehower, Addgene # 28106)[42] was also amplified and purified as described above, and nucleofected into parental astrocytes prior to experimentation. A NAPRT expression construct was purchased from GenScript (OHu28558D) and amplified and purified as described above. Plasmid was nucleofected in PPM1D$^{trnc.}$ astrocytes, selected with G418, and further isolated from single cell cultures.

**Western blotting.** Immunoblots were separated by SDS-PAGE and transferred to a PVDF membrane for analysis. All blots were blocked in 5% BSA (Gold Biotechnology) in 1X TBST (American Bio), and then were probed overnight at 4 °C, with primary antibodies raised against: PPM1D (SCBT F-10 sc-376257, 1:1000), GAPDH (Proteintech group HRP-60004, 1:5000), Actin (ThermoFisher MA5-11869, 1:2000), γH2AX pS139 (CST 2577, 1:1000), NAPRT (Proteintech group 66159-1-Ig, 1:2000), NAMPT (CST 86634, 1:1000), pCHK2 T68 (CST 2197, 1:1000), H3K27M (CST 74829, 1:1000), or p53 (CST 9282, 1:1000). Blots were then washed with 1X TBST and incubated with HRP conjugated- anti-mouse (ThermoFisher 31432, 1:10,000) or anti-rabbit (ThermoFisher 31462, 1:10,000) secondary antibodies for 1 h at room temperature (RT). Immunoblot exposure was carried out using Clarity Western ECL substrate (BioRad), and imaged on a ChemiDoc (BioRad) imaging system. Uncropped and unprocessed scans of all western blots shown are available in the Source Data file.

**In vitro chemical and IR treatments.** PPM1D$^{trnc.}$ astrocytes were treated with 50 μg/mL cycloheximide or 10 μM MG132 (both Sigma) for the indicated amount of time. Cells were then washed, pelleted, and lyzed for subsequent immunoblotting approaches, as described above. Quantification of immunoblot intensity was calculated using ImageJ software, and consisted of multiple ($n = 3$) blots. Irradiation of cells was performed using an X-RAD KV irradiator (Precision X-ray), and treatment consisted of an unfractionated, 10 Gy dose. PPM1D inhibitor treatment with GSK2830371 (Selleckchem), consisted of 50 nM treatment, 24 h prior to IR. FK866 (Selleckchem), GPP78 (Tocris Bioscience), STF118804 (Tocris Bioscience), STF31 (Tocris Bioscience), 5-azacytidine (Selleckchem), and DCT (Selleckchem) were dissolved in DMSO and used for treatment as indicated. Nicotinamide riboside (ChromaDex Inc.) and nicotinamide (Sigma) were dissolved in water while nicotinic acid (Sigma) was dissolved in PBS, prior to treatment alone or in combination with FK866, as indicated.

**γH2AX foci staining and imaging.** Astrocyte cell lines were seeded and incubated overnight, before radiation. Plates were then collected at indicated time points, fixed, permeabilized/blocked, and stained with primary and secondary antibodies for fluorescent imaging. Fixation was achieved with a 20 min RT incubation in fixation buffer (4% paraformaldehyde and 0.02% TritonX100, in PBS). Cells were subsequently washed in 1X PBS, followed by a joint permeabilization and blocking step in incubation buffer (5% BSA and 0.5% TritonX100, in PBS) for 1 h. Primary antibody raised against γH2AX pS139 (Millipore 05-636) was added at a dilution of 1:1000 in incubation buffer, and incubated overnight at 4 °C. Plates were washed, followed by a 1 h RT incubation with alexafluor-conjugated secondary antibodies (ThermoFisher A21425 or A11029, 1:10,000) and a nuclear dye, 1 μg/mL Hoechst 33342 (Sigma), in secondary buffer (0.5% TritonX100, in PBS). Plates were again washed, and imaged in PBS using the Cytation3 imaging system (BioTek). Images were stitched using Gen5 v2.09 software (BioTek), and both foci and cell numbers were counted using CellProfiler image processing software[43].

**Drug screen and cellular viability measurements.** In vitro cellular viability assessments of immortalized human astrocytes, MCF7, and U2OS cell lines were made using a previously described[13], high-content, microscopy platform developed by our group. In brief, cells were plated in a 96-well plate at a density of 2000 cells/well, and incubated overnight. Drug treatment or vehicle (0.5% DMSO) control was administered, and cells were incubated for 72-96 h as indicated. Plates were then washed with PBS and fixed with ice-cold 70% ethanol for 2 h at 4 °C. After removal of ethanol, plates were again washed with PBS, and stained for 30 min at RT, with 1 μg/mL Hoechst 33342 (Sigma). Cells were imaged using a Cytation3 imager (BioTek), and images were stitched and analyzed as described above. Viability assessments were made comparing drug treated to vehicle treated conditions. SU-DIPG and HSJD-DIPG-007 spheroid viability was assessed using CytoTox-Glo (Promega),

according to the manufacturer's protocols. Spheroids were treated with FK866 for 120 h before analysis using this method. IC$_{50}$ calculations were made using GraphPad Prism, by fitting data to an [inhibitor] vs response -variable slope four parameter nonlinear regression (as depicted in the representative figures).

**siRNA transfection and viability analysis.** Individual NAPRT targeting siRNAs were ordered from Dharmacon Inc. (Horizon Discovery), with target sequences listed in Supplementary Table 2. The panel of siRNAs used for synthetic lethal viability screening was hand-selected and ordered from Dharmacon Inc. and were provided in ON-TARGET plus mixtures, each containing up to four unique siRNAs per gene. $2 \times 10^5$ astrocytes were reverse-transfected with different siRNAs (200 nM final concentration), using Lipofectamine RNAiMAX (Invitrogen), according to manufacturer's protocols. For individual siRNAs, cells were incubated for 72 h, pelleted, and lyzed for immunoblotting. For the siRNA screen, cells were incubated for 24 h and split to different condition plates, where they were incubated for an additional 24 h. Cells were then treated with the described doses of FK866, and viability was assessed after 72 h of drug treatment, using the image-based pipeline described above. Viability measurements were made for each siRNA, and normalized to FK866-untreated conditions.

**NAD metabolome quantification.** The NAD metabolome was quantitatively analyzed using LC-MS/MS, using two separations on Hypercarb and 13C metabolite standards[16,44]. Subsequent NAD level analyses were performed using a NAD/NADH Quantification kit (Sigma), as per the manufacturer's specifications.

**Me/hME-DIP, bisulfite conversion, and global 5-hmC detection.** Genomic DNA was purified using the Wizard Genomic DNA purification kit (Promega), and subsequently immunoprecipitated or bisulfite-converted. Immunoprecipitation assays were performed using Me-DIP and hMe-DIP kits (Active Motif), according to suggested protocols. Immunoprecipitated DNA was extracted with phenol/chloroform and analyzed using quantitative PCR (qPCR), as described below. Bisulfite conversion was performed via EpiMark Bisulfite Conversion kit (NEB). Modified DNA was then amplified using EpiMark Hot Start Taq DNA polymerase (NEB), with primers listed in Supplementary Table 2, and purified with a PCR purification kit (Qiagen). Methylation was then assessed through Sanger-sequencing of the NAPRT promoter. Global 5-hmC levels were assessed via the Global 5-hmC quantification kit (Active Motif), according to manufacturer's protocols.

**Quantitative PCR (qPCR).** mRNA transcripts were purified from cells using a RNAeasy kit (Qiagen) and subsequently reverse transcribed using a High Capacity cDNA reverse transcription kit (Applied Biosystems). PPM1D and NAPRT gene expression levels were assessed through qPCR with TaqMan fluorescent probes (all from Applied Biosystems): PPM1D (4331182), NAPRT (4351372), and Actin (4333762F), according to manufacturer's protocol. Expression level fold change was calculated via ΔΔCt comparison, using Actin as a reference gene. The NAPRT promoter region was quantitated via qPCR using Fast Start Universal SYBR Green Master with ROX (Roche), and primers listed in Supplementary Table 2. All qPCR reactions were run on a StepOnePlus Real Time PCR system (Applied Biosystems).

**Infinium Methylation EPIC array and analysis.** 50–500 ng of genomic DNA was bisulfite-converted and analyzed for genome-wide methylation patterns using the Illumina Human EPIC Bead Array (850k) platform at the Yale Center for Genome Analysis (Astrocyte and SU-DIPG lines) or University College London Genomics Centre (HSJD-DIPG lines), according the manufacturer's instructions. Data was processed and analyzed using Genome Studio v1.9 for NAPRT specific probes and methylation β-values were generated for all probes for downstream analyses. Global hypermethylation assessments were made using Limma R package of t-test study, with false discovery correction (FDR) and an absolute β-values threshold, to identify probes that reached significance in methylation differential between PPM1D mutant and wild samples (also known as significantly variable probes, or SVPs). Top 2% most variable probes lists were selected for based on variance and analyzed from the dataset, as described above, filtered for CpG island probes and delta β 0.2, and used for comparison to publicly available data found in ref. [30], which was processed similarly.

**Chromatin immunoprecipitation (ChIP).** ChIP assays were performed using ChIP-IT Express kit (Active Motif), with a Rabbit IgG antibody (CST 2729) as an enrichment control. qPCR analysis for the NAPRT promoter was performed as described above. ChIP antibodies used: H3K4me1 (Abcam ab8895), H3K4me3 (CST 9751), H3K27me3 (CST 9733), and H3K27ac (Abcam, ab4729) at the manufacturer's recommended dilutions for ChIP.

**Animal handling and in vivo studies.** All animal use was in accordance with the guidelines of the Animal Care and Use Committee (IACUC) of Yale University and conformed to the recommendations in the Guide for the Care and Use of Laboratory Animals (Institute of Laboratory Animal Resources, National Research Council, National Academy of Sciences, 1996). Astrocyte xenograft studies were performed in NOD scid gamma (NSG, NOD.Cg-Prkdc$^{scid}$ Il2rg$^{tm1Wjl}$/SzJ female mice 3–4 weeks old) mice, and the protocol was approved by Yale University

IACUC. For cell line xenografts, $5 \times 10^6$ WT or PPM1D$^{trnc.}$ astrocytes, stably expressing firefly luciferase (lentivirus-plasmids from Cellomics Technology; PLV-10003), were combined with Matrigel (Corning, 47743-722) in a total volume of 0.2 mL. Cell-Matrigel suspension was injected subcutaneously into both the right and left flanks of shaved NSG mice. Mice were randomly sorted into treatment groups, and tumor burden and growth were measured on a weekly basis, via BLI intensity, as described below. FK866 was solubilized in DMSO at a concentration of 80 mg/ml. Mice were then administered the drug intraperitoneally twice a day for 4 days, repeated weekly for 3 weeks at 20 mg/kg in 10% cyclodextrin. Treatment began after 1 month of growth. Tumors were harvested after completion of treatment, and mass for each tumor was measured. Serially transplanted xenografts were created via continuous transplantation of PPM1D$^{trnc.}$ cell line xenografts in NSG mice. Subcutaneous flank injection with $5 \times 10^6$ cells was performed with Matrigel as described above. Mice were sorted randomly into treatment groups, and tumor volume was measured using standard caliper-based techniques. Tumor volume was calculated as length × width$^2$ × 0.52. U2OS xenograft studies were performed in athymic nude mice, also approved by Yale University IACUC. $5 \times 10^6$ cells were injected subcutaneously into the right flank of each animal and allowed to grow for 18 days before treatment. Mice were sorted randomly into treatment groups, and tumor burden was assessed through caliper measurement and volume calculations. FK866 was prepared and dosed as described above.

**Bioluminescent imaging of tumor burden**. Bioluminescence imaging (BLI) was performed using the IVIS Spectrum In Vivo Imaging System (PerkinElmer) according to the manufacturer's protocol. Images were taken on a weekly basis, and acquired 15 min post intraperitoneal injection with d-luciferin (150 mg/kg of animal mass). Quantification of BLI flux (photons/sec) was made through the identification of a region of interest (ROI) for each tumor, which was then circumscribed, background-corrected, and measured for BLI signal. Both right and left flank tumors were averaged together for each mouse, and then subsequently used for treatment group comparisons and analysis. All representative bioluminescent images were generated using a standard luminescent scale, and cropped to eliminate background objects.

**DIPG expression data**. Data from the Pacific Pediatric Neuro-Oncology Consortium (PNOC) NCT02274987 study[32] was kindly provided by The Genomics Research Institute (TGen), and contained *PPM1D* and *NAPRT* expression levels from 29 newly diagnosed DIPG cases. RNA-sequencing was performed using Illumina HiSeq per the manufacturer's protocol, and was used to calculate transcript abundance. Pearson's Correlation r was calculated using GraphPad Prism. Data from HSJD-DIPG lines and additional DIPG model cell lines was acquired from a previously published dataset which was collated from Affymetrix Agilent and Illumina expression arrays and from RNASeq and is described in ref. [28].

**Statistical analysis and significance**. Unless otherwise described, data was analyzed on Microsoft Excel and GraphPad Prism software. Student's two-tailed *T* test for significance was used for comparisons between two groups and described as significant at $^*p < 0.05$, $^{**}p < 0.01$, $^{***}p < 0.001$, $^{****}p < 0.0001$. Mann–Whitney test was used to assess tumor growth curves, using the same significance denotations as above. Log rank (Mantel-Cox) test was used to assess significance in tumor delay as measured by Kaplan–Meier plot. All error bars shown are standard deviation of the mean, unless indicated otherwise.

**Reporting summary**. Further information on research design is available in the Nature Research Reporting Summary linked to this article.

## Data availability
The methylation array data is available from the NCBI GEO database under the accession code GSE134165 [https://www.ncbi.nlm.nih.gov/geo/]. Expression data for HSJD-DIPG lines and additional DIPG model cell lines is described in ref. [28] and is available from the PedcBioPortal for Childhood Cancer genomics database [https://pedcbioportal.org]. All the other data supporting the findings of this study are available within the article and its supplementary information files and from the corresponding author upon reasonable request. A reporting summary for this article is available as a Supplementary Information file. Cell lines generated for this paper are available upon request.

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

## Acknowledgements

The authors would like to thank Drs. Timothy Chan, Ryan Jensen, Peter Glazer, and Michelle Monje for cell lines and reagents. This work was supported by funding from the NIH (5RO1CA215453-02 to R.S.B.), the American Cancer Society (128352-RSG-15-197-01 to R.S.B.), the Yale Cancer Biology Training Program via the Yale Cancer Center and Yale School of Medicine (N.R.F.), and philanthropic support from the Hope Through Hollis Fund in the TGen Foundation, the Whatever It Takes Foundation, the Team Cozzi Foundation (S.P. and M.E.B.), and the Roy J. Carver Trust (CB).

## Author contributions

Project was conceptualized by N.R.F. and R.S.B. Experimental design was performed by: N.R.F., R.K.S., G.A.B., C.B. and R.S.B. Experiments were carried out by N.R.F., R.K.S., R.L.M., M.S., A.N.K, and D.M.C. Project was supervised by: N.R.F., C.B., M.E.B., J.N., C.J, A.M.C. and R.S.B. Data was analyzed by N.R.F., G.A.B., S.P., M.S., A.M. and C.B. Manuscript was prepared by N.R.F. and R.S.B. All authors discussed results and commented on the prepared manuscript.

## Additional information

**Competing interests:** C.B. is the inventor of intellectual property on nicotinamide riboside, which was developed by ChromaDex Inc. C.B. owns equity in ChromaDex and serves as their chief scientific advisor. C.B. is on the scientific advisory board of ChromaDex and Cytokinetics. All remaining authors declare no competing interests.

