## [Peer Review File · Nature Communications]

Reviewers' Comments:

Reviewer #1:

Remarks to the Author:

In this manuscript by Fons et al., the authors sought to build novel isogenic astrocyte models of PPM1D mutant pediatric glioma for use in a drug screening platform to identify PPM1D mutation linked synthetic lethal targets. The authors successfully electroporate immortalized astrocytes with Cas9 and sgRNA targeting PPM1D exon 6 using a Lonza nucleofector and derive single cell clones with knock-in of the PPM1D truncating mutation seen in several DIPG cancers. They then use this model for a paired drug screen identifying drugs with significantly lower IC50 in knock-in clones vs. parental cells. Interestingly, they identify that the knock-in clones with truncating PPM1D mutations have a sensitivity to NAMPT inhibitor FK866 at the low nM range of dosing that is not present in parental or unedited clone controls. The authors then provide additional experimental evidence using multiple NAMPT inhibitor compounds and an over-expression model to show that this sensitivity is likely the result of higher PPM1D activity, not necessarily a neomorphic function of the truncating mutation that induces this sensitivity. The manuscript is generally well written although more detail is needed, and for accurate evaluation of the work. Critiques are addressed below as major and minor points:

Major Points:

1) Figure 3.E and describing text in the results section on page 7. It is unclear how a determination of CIMP was made because the annotation of the probes chosen for clustering was poorly described. The 850K array has many probes outside of CpG islands so were the SVs described only the CpG island probes or were other probes considered. Furthermore, the description of steps used to calculate the methylation clustering was not described in sufficient detail to allow for critical comment as it is unclear whether Z-score, M-value, or Beta value was utilized in the derivation of significantly variable probes and what was used for clustering. Overall, there is just too little detail to be able to evaluate what analyses the authors actually performed and how cogent those methods were. In the case that variability was assessed at the level of Beta values, plotting as in (Noushmehr et al., 2010)[Figure 1B] of raw Beta values and clustering is likely also warranted instead of Z-score plotting which may magnify small effects.

2) Please provide more balance to the discussion, for while it's true that you provide at least one additional model, HCT116, being sensitive to FK-866 inhibitor sensitivity a cursory look at the cancer dependency map <https://depmap.org> shows no particular trend for PPM1D mutant lines to be dependent on NAMPT or sensitive to FK-866. In defense of your work there is at least one PPM1D truncating mutant line that shows profound sensitivity to NAMPT inhibition and FK-866 and resistance could be due to other mutations. I make this critique because it's a long way to argue that a single cell line from another cancer that shows sensitivity to FK-866 indicates generalizability or that the expression correlations detected in low grade glioma, breast carcinoma, and ovarian serous cystadenocarcinoma will functionally result in sensitivity.

3) The major findings of this study were derived from the isogenic astrocyte model system. Unfortunately, this isogenic astrocyte model system is not a faithful recapitulation of PPM1D mutations in brainstem glioma because the authors did not include H3.3K27M mutation in this model system. It has been observed that most PPM1D mutations co-occur with the H3.3K27M mutation in brainstem gliomas¹ and the dominant-negative effect of H3.3K27M mutant could lead to a global hypomethylation phenotype¹. To improve the clinical significance of this study, it is therefore important to ensure the critical findings (CIMP phenotype, NAMPT deficiency, and NAMPT inhibitor sensitivity) induced by PPM1D truncating mutation is also truly exist in the context of H3.3K27M mutation in this isogenic astrocyte model system.

4) The authors need to clearly describe how the NHA cell line was immortalized in the manuscript. In Dr. Pieper's original paper, the authors used different combinations of transgenes, such as hTERT, hTERT/E7, hTERT/E6/E7, and hTERT/E6/E7/Ras to immortalize NHA cells². If the immortalized NHA cell line used in this study expresses E6 protein, which can inactivate p53 function, this might raise another concern about the functional redundancy of PPM1D mutations and E6 expression in suppressing p53 pathway in this isogenic astrocyte model.

5) For Fig.2H, the authors show that one PPM1D-mutant DIPG cell line (SU-DIPG35) does not

express NAPRT. As NAPRT deficiency has been observed in many cancer cell lines and in a large percentage of tumor tissues of small cell lung carcinomas, GBM and oligodendrogliomas³, in which the PPM1D mutations are infrequent, it is therefore very speculative to conclude that PPM1D mutation is the cause of NAPRT deficiency based on the data from one DIPG cell line. Can the authors validate this finding in other PPM1D-mutant DIPG cell lines?

6) For Fig. 4C, the authors analyzed the correlation of PPM1D expression and NAPRT expression in a cohort of DIPG biopsy specimens. Since PPM1D truncating mutation has a much stronger effect than PPM1D overexpression in terms of increasing total PPM1D protein level and phosphatase activity, it would be more appropriate to analyze the correlation of PPM1D mutation status and NAPRT expression for Fig. 4C.

Minor Points:

1) Please label supplementary figure S1 with the putative frameshift codon induced in the single cell clones by knock-in. Perhaps I missed it but I could not find the annotation for what the putative knock-in amino acid site was for the astrocytes even though the figure legend of Figure 1A mentions it is colored in red... Was it at R458 as the overexpression used or another site?

2) Please state the genotype of immortalized astrocytes used in the study and provided by Timothy Chan. e.g. hTERT, or hTERT + E6, or hTERT + E6/E7.

3) The red arrows are missing in Fig. 1A.

4) For Fig. 1E, a scale bar is needed.

5) The method used for IC50 calculation need to be described in the manuscript.

6) The authors need to list all the compounds used in the initial screening in the supplemental table.

7) For Figs.2B and 3E, it would be better to recolor the heat maps using green-magenta instead of red-green combination.

8) Why the SU-DIPG#4 cell line (Fig. 2H) was not included in the NAMPT inhibitor test (Fig. 2I)?

9) In Fig. S6A, the authors observed that FK866 significantly inhibited the tumor growth of HCT116. Although HCT116 harbors PPM1D truncating mutation, this result is still not informative as no isogenic control cell line was included in this experiment to demonstrate the tumor-specific cell killing of FK866. Moreover, previous studies have reported that HCT116 also expressed NAPRT³⁴. This raises a concern whether HCT116 is a good model to study the correlations of PPM1D mutation and NAPRT deficiency.

10) Previous studies in Nature Genetics did not observe an obvious CIMP phenotype associated with PPM1D mutations in brainstem glioma¹. The authors need to address this kind of inconsistency in the discussion.

References

1. Zhang, L. et al. Exome sequencing identifies somatic gain-of-function PPM1D mutations in brainstem gliomas. *Nat. Genet.* 46, 726–730 (2014).
2. Sonoda, Y. et al. Formation of intracranial tumors by genetically modified human astrocytes defines four pathways critical in the development of human anaplastic astrocytoma. *Cancer Res.* 61, 4956–4960 (2001).
3. Cole, J. et al. Novel NAPRT specific antibody identifies small cell lung cancer and neuronal cancers as promising clinical indications for a NAMPT inhibitor/niacin co-administration strategy. *Oncotarget* 8, 77846–77859 (2017).
4. Tateishi, K. et al. Extreme Vulnerability of IDH1 Mutant Cancers to NAD⁺ Depletion. *Cancer Cell* 28, 773–784 (2015).
5. Noushmehr, H., Weisenberger, D. J., Diefes, K., Phillips, H. S., Pujara, K., Berman, B. P., . . . Aldape, K. (2010). Identification of a CpG Island Methylator Phenotype that Defines a Distinct Subgroup of Glioma. *Cancer Cell*, 17(5), 510-522. doi: <http://dx.doi.org/10.1016/j.ccr.2010.03.017>

Reviewer #2:

Remarks to the Author:

NCOMMS-18-34272 by Fons et al. (Bindra)

“PPM1D Mutations Silence NAPRT Gene Expression and Confer Exquisite NAMPT Inhibitor Sensitivity in Glioma”

Brief summary

In this article, the authors sought to identify the therapeutic vulnerabilities that are caused by PPM1D mutations in glioma. To achieve this, they performed chemical inhibitor and siRNA screens. They found that clinically identified truncation mutations in PPM1D lead to enhanced phosphatase activity of PPM1D, due to the increased stability of PPM1D protein. The authors further observed that PPM1D mutations result in decreased expression of NAPRT by regulating DNA methylation, causing increased sensitivity to the NAMPT inhibitors in vitro and in vivo.

Strengths and weaknesses

Strengths: The authors performed several unbiased screens to identify and characterize the mechanisms underlying PPM1D mutation-mediated therapeutic vulnerability. They show that PPM1D mutant cells are sensitive to NAMPT inhibitors. This study identified a novel therapeutic strategy for pediatric gliomas and even other solid tumors with PPM1D mutations, that are extremely lethal with poor prognosis.

Weakness: The manuscript lacks mechanistic novelty. The authors should provide more in-depth mechanistic studies to strengthen their hypothesis.

Additional comments

1. In Figure 1J, the results would be more convincing if the authors included a phosphatase dead PPM1D mutant protein overexpression as a negative control.
2. In Figure 2E, the authors show that NR treatment can bypass the cytotoxicity of NAMPT inhibition. What is the outcome when the cells are treated with other NAD⁺-precursors, such as NA and NAM?
3. Although the authors conclude that PPM1D mutations affect NAPRT expression due to DNA hypermethylation, they also show that PPM1D mutant cells have lower H3K4me3 and H3K27ac marks at the promoter of the NAPRT gene. Can the expression of NAPRT be rescued with a DNA methylation inhibitor, such as 5-Aza?
4. Mechanistically, the authors show that PPM1D mutation induces silencing of the NAPRT gene by regulating DNA methylation at its promoter. Since TET mutation is also correlated with a wide range of different cancers, is there mechanistic correlation between PPM1D mutation with TET activity in regulating DNA methylation at the promoter of NAPRT? What is the potential mechanism driving the DNA methylation by mutated PPM1D?
5. Can treating the mice with NAD⁺ precursors such as NR or NA alter the in vivo sensitivity to FK866 in PPM1D mutant xenograft tumors?
6. The authors show similar results with HCT116 cell line xenografts in vivo. To draw a conclusion that it is a common mechanism in PPM1D mutation tumors, the authors should also provide the evidence showing the mutant PPM1D-induced NAPRT deficiency in HCT116 cells or other cell line harboring PPM1D mutation in vitro.

Fons et al 2019, Nature Communications

We would like to take a moment to thank both reviewers for their thoughtful review and insightful comments regarding our manuscript. We are pleased that both reviewers acknowledge the significance, novelty, and major clinical implications of our recent discovery that *PPM1D* mutations induce NAPRT silencing and confer sensitivity to NAMPT inhibitors. In particular, as noted by Reviewer 2, “this study identified a novel therapeutic strategy for pediatric gliomas and even other solid tumors with *PPM1D* mutations, that are extremely lethal with [a] poor prognosis.”

The primary concern raised by both reviewers was whether we can validate mutant *PPM1D*-induced CIMP, NAPRT silencing, and NAMPT inhibitor sensitivity in additional patient-derived, *PPM1D*-mutant DIPG cell line models, as well as in tumor cell lines with endogenous *PPM1D* alterations derived from other solid tumor types. In direct response to this request, we have performed experiments in 3 additional patient-derived DIPG models, and 2 additional established osteosarcoma and breast cancer cell lines with either truncated or amplified *PPM1D*. There is a dearth of patient-derived DIPG models available in the world, but we were able to obtain additional *PPM1D*-mutant DIPG cell lines from the laboratory of Dr. Angel Carcaboso (Developmental Tumor Biology Laboratory, Hospital San Joan de Deu, Barcelona, Spain), which were subjected to whole genome methylation profiling in the laboratory of Dr. Chris Jones (ICR, London, England, United Kingdom; both are now co-authors in the revised manuscript). We are pleased to report that we have confirmed and validated the link between mutant *PPM1D*, CIMP, and *NAPRT* promoter methylation in these additional cell lines. Specifically, in the revised manuscript, we have now demonstrated mutant *PPM1D*-induced or -associated CIMP, NAPRT silencing and/or NAMPT inhibitor sensitivity in: (a) immortalized astrocytes with CRISPR/Cas-engineered heterozygous *PPM1D* truncating mutations, (b) immortalized astrocytes stably overexpressing WT or mutant *PPM1D* ORFs, (c) 4 unique primary, patient-derived *PPM1D* mutant glioma models, and (d) two cell lines with endogenous *PPM1D* aberrations (in total, 9 unique model cell lines; not including several isogenic and matched cell lines with WT *PPM1D*). These studies further strengthen the link between mutant *PPM1D*, CIMP, NAPRT silencing, and NAMPT inhibitor sensitivity. Coupled with our extensive mechanism of action studies described in the initial manuscript, which implicate NAPRT status as the driver of NAMPT inhibitor sensitivity in *PPM1D*-mutant cancers, these data further corroborate this interaction, as well as the significance of our findings to multiple solid tumor types.

Both reviewers also requested several additional experiments to further clarify possible mechanism(s) of action, experimental methodology, and also the clinical relevance of our work, including: (a) the methodology that we employed to assess CIMP; (b) potential effects of H3.3 K27M mutations; (c) a discussion regarding why our findings were not discovered by others, during the course of previously published studies on *PPM1D*-mutant glioma; (d) confirmation that mutant *PPM1D*-induced NAMPT inhibitor sensitivity is dependent on elevated levels of *PPM1D* phosphatase activity; (e) whether NAD precursors can rescue NAMPT inhibitor cytotoxicity; (f) assessment of potential CIMP mechanisms and mediators, including TET proteins, and whether methylation inhibitors can reverse the phenotype; and (g) demonstration of mutant *PPM1D*-associated NAMPT inhibitor sensitivity in another cell line model *in vivo*. We successfully performed several new experiments to address each of these important suggestions proposed by the two reviewers. Collectively, these new data further strengthen the link between mutant *PPM1D*, CIMP, NAPRT silencing, and NAMPT inhibitor sensitivity, in both glioma and other solid tumor types.

Below is a point-by-point response to the comments from both Reviewers.

Response to Reviewer 1 comments:

A. Major Points

1. *“It is unclear how a determination of CIMP was made because the annotation of the probes chosen for clustering was poorly described. The 850K array has many probes outside of CpG islands so were the SVPs described only the CpG island probes or were other probes considered. Furthermore, the description of steps used to calculate the methylation clustering was not described in sufficient detail to allow for critical comment as it is unclear whether Z-score, M-value, or Beta value was utilized in the derivation of significantly variable probes and what was used for clustering. Overall, there is just too little detail to be able to evaluate what analyses the authors actually performed and how cogent those methods were. In the case that variability was assessed at the level of Beta values, plotting as in (Noushmehr et al., 2010) [Figure 1B] of raw Beta values and clustering is likely also warranted instead of Z-score plotting which may magnify small effects.”*

This is an excellent point, regarding the definition of CIMP, and how it was assessed from our methylation data of astrocyte and DIPG cell lines. While definitions of CIMP are far ranging, we chose to use a definition seen in glioma CIMP (also known as G-CIMP), as described in work by Turcan *et al* for IDH1/2-mutant gliomas [1]. In direct response to this comment, we have provided detailed methodology regarding our approach to CIMP classification in the *Methods* section of the revised manuscript. In addition, as suggested by the reviewer, we have re-analyzed the data using raw beta values, instead of the previously reported z-scores, in order to provide a more consistent determination of CIMP (**Figure 3e** in the revised manuscript).

2. *“Please provide more balance to the discussion, for while it’s true that you provide at least one additional model, HCT116, being sensitive to FK-866 inhibitor sensitivity a cursory look at the cancer dependency map <https://depmap.org> shows no particular trend for PPM1D mutant lines to be dependent on NAMPT or sensitive to FK-866. In defense of your work there is at least one PPM1D truncating mutant line that shows profound sensitivity to NAMPT inhibition and FK-866 and resistance could be due to other mutations. I make this critique because it’s a long way to argue that a single cell line from another cancer that shows sensitivity to FK-866 indicates generalizability or that the expression correlations detected in low grade glioma, breast carcinoma, and ovarian serous cystadenocarcinoma will functionally result in sensitivity.”*

This is an important point regarding the applicability of our findings to other tumor types, since our novel discovery of mutant PPM1D-induced CIMP, NAPRT silencing and NAMTP inhibitor sensitivity was based on a relatively small number of cell lines, in the initial version of the manuscript. While we found that NAPRT gene expression data from CBioPortal do maintain a statistically significant difference between PPM1D-low and -high expressing low grade gliomas, breast, and ovarian cancers, we do recognize the limitations of this type of large data analysis. As such, the Reviewer appropriately requests a more balanced discussion regarding the relevance of our discovery to multiple tumor types. In the initial version of the manuscript, we reported data with several PPM1D-WT and -mutant cell line models, including: (1) isogenic, PPM1D-mutant and WT immortalized astrocytes engineered to harbor several unique, PPM1D truncating mutations, as well as those expressing both WT and mutant open reading frames (ORFs); and (2) a panel of three WT and one *PPM1D* mutant patient-derived DIPG neurosphere cultures provided by the laboratory of Dr. Michele Monje (Stanford School of Medicine, Stanford, CA).

To directly address these concerns above, we have tested several additional cell lines to support the generalizability of mutant PPM1D-induced CIMP, NAPRT silencing, and NAMPT inhibitor sensitivity, in additional patient-derived and established cancer cell lines, derived from a diverse range of tumor tissue types. There is a paucity of *PPM1D*-mutant, DIPG patient-derived cell lines available in the research community. However, in response to the Reviewer's concerns, we were able to obtain data from three additional patient-derived, *PPM1D*-mutant DIPG lines from the laboratory of Dr. Angel Carcaboso (HSJD-DIPG-007, 008 and -014b; Developmental Tumor Biology Laboratory, Hospital San Joan de Deu, Barcelona, Spain), which were subjected to whole genome methylation profiling in the laboratory of Dr. Chris Jones (ICR, London, England, United Kingdom). We confirmed that each of these three lines demonstrated a global CIMP phenotype similar to that observed in our other *PPM1D*-mutant cell line models, relative to PPM1D-WT cells (revised **Fig. 3e**). Furthermore, we found that the *NAPRT* promoter is focally hypermethylated in HSJD-DIPG-007 and -008, again at levels similar to that which was seen in our previously tested immortalized astrocyte and patient-derived DIPG model cell lines (revised **Fig. 3f**). While HSJD-DIPG-014b cells clearly displayed a CIMP phenotype similar to other lines tested, we did not observe significant hypermethylation at the *NAPRT* promoter. This discrepancy between the level of CIMP that we observed and the lack of clear *NAPRT* promoter silencing in this particular cell line could be explained by one of many possible factors, including: (a) inadequate sensitivity to detect silencing at the *NAPRT* promoter, as these data only were based on selected probes from the ~850K CpG islands in the array, which are localized within the *NAPRT* promoter (e.g., higher resolution, bisulfite conversion and sequencing of the entire promoter may reveal evidence of promoter silencing); (b) differences in culture conditions that could lead to variability in methylation profiles; (c) intrinsic heterogeneity associated with this novel type of mutant PPM1D-induced CIMP (e.g., while we detected mutant PPM1D-associated CIMP in this line, the *NAPRT* promoter may not be specifically silenced in this tumor cell line subclone); or (d) other co-occurring mutations or molecular differences affecting the penetrance of mutant PPM1D-induced *NAPRT* silencing. However, these data confirm that we can detect mutant PPM1D-associated CIMP and *NAPRT* silencing in additional patient-derived DIPG cell lines, which directly addresses the requests and concerns raised by the reviewer here, and also below in point (5).

In parallel, we also identified additional cell lines with truncating PPM1D mutations or gene amplifications derived from other human tumor tissues, and we again tested whether we could see a similar phenotype. Based on previous observations, we identified a truncating PPM1D mutation in the osteosarcoma cell line, U2OS, and gene amplification in the breast cancer cell line, MCF7. As shown in the new **Supplementary Figs. 5a** and **5b**, we confirmed that these genetic aberrations were associated with high levels of PPM1D expression by western blot analysis and qPCR analysis of PPM1D mRNA levels; the latter assay was performed to confirm that the PPM1D gene amplification seen in MCF7 cells correlates with elevated PPM1D mRNA levels. In these two additional cell lines, elevated PPM1D expression correlated with undetectable levels of *NAPRT* protein and significantly reduced mRNA levels (new **Supplementary Fig. 5a** and **5c**, respectively). We then performed bisulfite conversion and sequencing of the *NAPRT* promoter in these cell lines, which confirmed silencing by promoter hypermethylation (new **Supplementary Fig. 5d**). Finally, we demonstrated that mutant PPM1D-associated *NAPRT* silencing was associated with significant NAMPT inhibitor sensitivity in both cell lines, at levels similar to those observed for the other *PPM1D*-mutant cell lines tested in the original manuscript (**Supplementary Fig. 5e**).

Taken together, we have performed studies on six additional cell line models with *PPM1D* aberrations, which include both primary tumor and established models, derived from a diverse range of human tumor tissue types. Specifically, in the revised manuscript we have now demonstrated mutant PPM1D-induced or -associated CIMP, *NAPRT* silencing, and/or NAMPT inhibitor sensitivity in: (a) immortalized

astrocytes with CRISPR/Cas-engineered heterozygous PPM1D truncating mutations, (b) immortalized astrocytes stably overexpressing WT or mutant PPM1D ORFs, (c) 4 unique primary, patient-derived *PPM1D* mutant models, and (d) two cell lines with endogenous *PPM1D* aberrations (in total, 9 unique model cell lines; not including several isogenic and matched cell lines with WT PPM1D genes). These studies further strengthen the link between mutant PPM1D, CIMP, NAPRT silencing, and NAMPT inhibitor sensitivity. Coupled with our extensive mechanism of action studies described in the initial manuscript, which implicate NAPRT status as the driver of NAMPT inhibitor sensitivity in *PPM1D*-mutant cancers, these data further corroborate this interaction, as well as the significance of our finding to multiple solid tumor types.

3. *“The major findings of this study were derived from the isogenic astrocyte model system. Unfortunately, this isogenic astrocyte model system is not a faithful recapitulation of PPM1D mutations in brainstem glioma because the authors did not include H3.3K27M mutation in this model system. It has been observed that most PPM1D mutations co-occur with the H3.3K27M mutation in brainstem gliomas1 and the dominant-negative effect of H3.3K27M mutant could lead to a global hypomethylation phenotype1. To improve the clinical significance of this study, it is therefore important to ensure the critical findings (CIMP phenotype, NAPRT deficiency, and NAMPT inhibitor sensitivity) induced by PPM1D truncating mutation is also truly exist in the context of H3.3K27M mutation in this isogenic astrocyte model system.”*

Reviewer 1 raises an excellent point regarding the possible impact of H3.3 K27M mutations, which often co-occur with *PPM1D* mutations in pediatric brainstem gliomas. These H3.3 mutations have been studied extensively, and are thought to induce DNA hypomethylation. However, these studies have been generally limited in scope with regard to the number of samples tested, and they have not always isolated possible confounding mutations (such as within PPM1D), in their analyses. As such, the reviewer requests that we demonstrate the detection of mutant PPM1D-induced CIMP, NAPRT silencing, and NAMPT inhibitor sensitivity in cell line models which also harbor H3.3 K27M mutations. We have included additional data in the revised manuscript which directly addresses this important concern. As shown in the new **Supplementary Fig. 6a**, we created a table of the DIPG cell lines tested along with PPM1D and histone 3 K27M mutation status, which includes the three additional DIPG patient-derived cell lines that were tested in the revised manuscript (HSJD-DIPG-007, 008 and -014b). As shown in this table, these cell lines were all reported to harbor H3.3 K27M mutations, and we confirmed the protein expression of the encoded mutant histone with an H3.3 K27M-mutant specific antibody, in a subset of these cell lines by western blot analysis (new **Supplementary Fig. 6b**). We further demonstrated CIMP in all four DIPG cell lines which harbor both PPM1D and H3.3 K27M mutations (SU-DIPG XXXV, HSJD-DIPG-007, 008 and -014b), which was not observed in the PPM1D-WT DIPG cell lines (which also harbor H3.1 or H3.3 K27M mutations; SU-DIPG-IV, -XIII and -XVII; revised **Figs. 3e** and **3f**, respectively), and NAPRT promoter methylation in 3 of them (SU-DIPG-XXXV, HSJD-DIPG-007, and -008). We were able to consistently passage the SU-DIPG cell lines for periods of time that were sufficient for reliable, longer term cell viability assays, which confirmed NAMPT inhibitor sensitivity specifically in the *PPM1D* mutant, but not WT cell lines (revised **Fig. 2j**). In the original manuscript, we reported mutant PPM1D-dependent NAMPT inhibitor sensitivity in immortalized astrocytes engineered to contain heterozygous truncating PPM1D mutations, as well as those which overexpress ORFs encoding either WT or mutant PPM1D, and these lines harbor WT H3.3 genes. In the revised manuscript, we have also confirmed that we can detect NAPRT silencing and NAMPT inhibitor sensitivity in two additional PPM1D-mutant cell lines, U2OS and MCF7, these lines harbor WT H3.3 genes (new **Supplementary Fig 5**). Taken together, these data strongly suggest that mutant PPM1D-induced NAPRT silencing occurs independent of H3.3 K27M mutation status. As noted by the reviewer, these findings greatly enhance the clinical significance of our

discovery, and they directly address his/her important concerns about the specific dependency of the *PPM1D* mutation in this novel interaction.

4. The authors need to clearly describe how the NHA cell line was immortalized in the manuscript. In Dr. Pieper's original paper, the authors used different combinations of transgenes, such as hTERT, hTERT/E7, hTERT/E6/E7, and hTERT/E6/E7/Ras to immortalize NHA cells². If the immortalized NHA cell line used in this study expresses E6 protein, which can inactivate p53 function, this might raise another concern about the functional redundancy of PPM1D mutations and E6 expression in suppressing p53 pathway in this isogenic astrocyte model.

The reviewer requests additional information regarding the origin of the parental astrocyte cell line that was utilized in our experiments. To address this concern, we have further stated within the methods section that these cell lines are “immortalized using hTERT/E6/E7”. The reviewer also raises a valid concern regarding the functional redundancy between E6 expression and *PPM1D* mutations, and whether it could serve as a confounding variable regarding the link between mutant *PPM1D* and NAPRT silencing. In the original manuscript, we demonstrated CIMP, NAPRT silencing, and NAMPT inhibitor sensitivity in the patient-derived DIPG model, SU-DIPG-XXXV, which is WT for p53 (revised **Fig. 2j** and please see new **Supplementary Fig. 6a**). In addition, as noted above, we also confirmed CIMP in three additional *PPM1D*-mutant, patient-derived DIPG cell lines in the revised manuscript, along with NAPRT promoter silencing in two of them (revised **Fig. 3e** and **3f**), which are also p53 WT (new **Supplementary Fig. 6a**). In addition, we confirmed p53 expression by western blot analysis for two of these *PPM1D*-mutant DIPG cell lines (i.e., SU-DIPG-XXXV and HSJD-DIPG-007; new **Supplementary Fig. 6b**). Finally, as mentioned above, we also demonstrated NAPRT silencing and NAMPT inhibitor sensitivity in two additional *PPM1D*-mutant cell lines, U2OS and MCF7, which are both WT for p53 (e.g., [2]; new **Supplementary Fig 5**). Collectively, these data indicate that mutant *PPM1D*-induced CIMP and NAPRT silencing occurs independent of p53 mutation status.

5. “For Fig.2H, the authors show that one PPM1D-mutant DIPG cell line (SU-DIPG35) does not express NAPRT. As NAPRT deficiency has been observed in many cancer cell lines and in a large percentage of tumor tissues of small cell lung carcinomas, GBM and oligodendrogliomas³, in which the PPM1D mutations are infrequent, it is therefore very speculative to conclude that PPM1D mutation is the cause of NAPRT deficiency based on the data from one DIPG cell line. Can the authors validate this finding in other PPM1D-mutant DIPG cell lines?”

This is an important point, and we have addressed this directly by demonstrating mutant *PPM1D*-induced CIMP, NAPRT silencing, and NAMPT inhibitor sensitivity using an expanded set of five unique cell lines (discussed in the response to point 2 above). In particular, we have demonstrated NAPRT silencing and/or CIMP in three additional *PPM1D*-mutant patient-derived DIPG models, which directly addresses the concern raised here. We agree with the reviewer that NAPRT expression is undetectable in a range of tumors, albeit infrequently, although the mechanisms by which the expression is downregulated are poorly understood. We believe that the work presented in this manuscript sheds critical insight into one potential mechanism by which NAPRT is silenced in selected tumors.

6. “For Fig. 4C, the authors analyzed the correlation of PPM1D expression and NAPRT expression in a cohort of DIPG biopsy specimens. Since PPM1D truncating mutation has a much stronger effect than PPM1D overexpression in terms of increasing total PPM1D protein level and phosphatase activity, it would be more appropriate to analyze the correlation of PPM1D mutation status and NAPRT expression for Fig. 4C.”

The reviewer requests a correlational analysis between NAPRT expression and *PPM1D* mutation status, instead of *PPM1D* expression. In response to this request, we have performed this analysis, which again reveals a general correlation between reduced NAPRT expression and mutant *PPM1D* tumors (see new **Fig. 4c and Supplementary Fig. 8d**). While it is correct that *PPM1D* truncation mutations typically increase cellular *PPM1D* protein levels, it should be noted that *PPM1D* amplification and/or overexpression can also induce *PPM1D* protein levels that are comparable to that observed with truncating mutations. This is exemplified by a comparison of *PPM1D* protein levels between MCF7 cells, which harbor a *PPM1D* amplification, and both U2OS and our engineered astrocyte cell lines, which harbor truncating *PPM1D* mutations (new **Supplemental Fig. 5a**). As such, we have also included the original *PPM1D* expression vs. NAPRT expression correlation figure in the new **Supplementary Fig. 8c**.

B. Minor Points

The reviewer makes several minor, but very important and thoughtful points regarding data presentation, clarifications about the data, and methodology. We have addressed each of these points in the revised manuscript, which are briefly summarized below.

1. *Please label supplementary figure S1 with the putative frameshift codon induced in the single cell clones by knock-in.* We have labelled this in the revised manuscript.
2. *Please state the genotype of immortalized astrocytes used in the study and provided by Timothy Chan. e.g. hTERT, or hTERT + E6, or hTERT + E6/E7.* We added this information in the revised manuscript.
3. *The red arrows are missing in Fig. 1A.* We have added this to the revised manuscript.
4. *For Fig. 1E, a scale bar is needed.* We have added this to the revised manuscript.
5. *The method used for IC50 calculation need to be described in the manuscript.* We have included this information in the revised manuscript.
6. *The authors need to list all the compounds used in the initial screening in the supplemental table.* We have created a table with this information in the revised manuscript (new **Supplementary Table 1**)
7. *For Figs.2B and 3E, it would be better to recolor the heat maps using green-magenta instead of red-green combination.* We have adjusted the color of the heat maps for enhanced interpretation in the revised manuscript.
8. *Why the SU-DIPG#4 cell line (Fig. 2H) was not included in the NAMPT inhibitor test (Fig. 2I)?* In direct response to this request, we have included cell viability data for SU-DIPG-IV with the NAMPT inhibitor, which is now shown in revised **Fig. 2j**.
9. *In Fig. S6A, the authors observed that FK866 significantly inhibited the tumor growth of HCT116. Although HCT116 harbors PPM1D truncating mutation, this result is still not informative as no isogenic control cell line was included in this experiment to demonstrate the tumor-specific cell killing of FK866...This raises a concern whether HCT116 is a good model to study the correlations of PPM1D mutation and NAPRT deficiency.* The reviewer raises an excellent concern regarding HCT116, as it harbors an endogenous *PPM1D* truncating mutation, and thus there is no isogenic control that is readily

available. Because of this issue, we initially focused on the immortalized astrocyte cell line models that we engineered to be isogenic for *PPM1D* aberrations (i.e., heterozygous truncating *PPM1D* mutations, as well as WT and mutant *PPM1D* ORFs). In parallel, we now have confirmed mutant *PPM1D*-associated CIMP, NAPRT silencing, and/or NAMPT inhibitor sensitivity in 9 unique model cell lines. In direct response to this concern, we have removed data obtained with the HCT116 cell line from the revised manuscript.

10. *Previous studies in Nature Genetics did not observe an obvious CIMP phenotype associated with PPM1D mutations in brainstem glioma. The authors need to address this kind of inconsistency in the discussion.* The reviewer makes an excellent point that a previous study in Nature Genetics by Zhang *et al*, which described the presence of *PPM1D* mutations in brainstem gliomas, did not detect mutant *PPM1D*-induced CIMP [3]. This study was focused on characterizing the genetic landscape of gliomas arising locations within the brainstem and thalamus in pediatric and adult patients, and truncating *PPM1D* mutations were unexpectedly identified. In parallel, Wu *et al*. also reported *PPM1D* mutations in a panel of pediatric high-grade gliomas, including DIPGs and non-brainstem gliomas [4]. Taylor *et al* also reported similar mutations in a study of DIPGs, which was also published in 2014 [5]. In each of these studies, only a handful of tumors with *PPM1D* mutations were studied, ranging between 5-7 cases, within sample sets ranging from 26-127 cases in total. While Zhang *et al* performed whole genome methylation profiling in their study, the samples were subjected to unsupervised hierarchical clustering, consensus clustering, and principal component analysis for the 2% most variant loci. Only five *PPM1D*-mutant cases were included within a large of group of tumors, and a significant of number IDH1-mutant tumors were included, thus, it is likely that this particular study was not designed to detect a mutant *PPM1D*-induced CIMP phenotype. In the other two studies, whole genome methylation was not performed. In response to the reviewer's request, we have included this point in the discussion.

Response to Reviewer 2 comments:

1. *"In Figure 1J, the results would be more convincing if the authors included a phosphatase dead PPM1D mutant protein overexpression as a negative control."*

Reviewer 2 makes an excellent suggestion that we test a phosphatase-dead *PPM1D* (D314A) expression construct as a negative control to demonstrate that NAMPT inhibitor sensitivity is dependent on increased *PPM1D* phosphatase activity. In direct response to this suggestion, we acquired a phosphatase-dead, *PPM1D* D314A mammalian expression vector and then tested whether the expression of this protein conferred NAMPT inhibitor sensitivity. As expected, we found that these cells were resistant to FK866, suggesting that *PPM1D* phosphatase activity is required for the induction of NAMPT inhibitor sensitivity in *PPM1D*-mutant cells. We have included this data in the revised manuscript (revised **Fig. 1j**).

2. *"In Figure 2E, the authors show that NR treatment can bypass the cytotoxicity of NAMPT inhibition. What is the outcome when the cells are treated with other NAD⁺-precursors, such as NA and NAM?"*

Reviewer 2 requests that we test whether NA and NAM also can reverse NAMPT inhibitor sensitivity in *PPM1D*-mutant cells. We have performed these experiments, which reveal that (a) exogenously added NAM can out-compete FK866 at the active site of NAMPT, and rescue mutant *PPM1D*-induced NAMPT inhibitor sensitivity, while (b) NA is unable to rescue this effect *in vitro*. As shown in the schematic in the revised **Fig. 2a**, NAM or NR can rescue NAMPT inhibitor toxicity, by increasing enzyme substrate or bypassing the enzyme entirely, respectively, to create NAD (i.e., independent of NAPRT status).

However, NA must be converted to NAMN by NAPRT before it can be converted to NAD in the Preiss-Handler pathway, and we have demonstrated that this gene is silenced in PPM1D-mutant cells. These points are further discussed in point (5) below. Collectively, these results further suggest that NAPRT silencing drives mutant PPM1D-induced NAMPT inhibitor sensitivity. We have included this data in the new **Supplementary Fig. 2g-l**.

3. *“Although the authors conclude that PPM1D mutations affect NAPRT expression due to DNA hypermethylation, they also show that PPM1D mutant cells have lower H3K4me3 and H3K27ac marks at the promoter of the NAPRT gene. Can the expression of NAPRT be rescued with a DNA methylation inhibitor, such as 5-Aza?”*

The reviewer asks for further characterization of the DNA methylation patterns found at the *NAPRT* promoter. Specifically, he/she asks whether we can reverse promoter methylation by treatment with demethylating agents, such 5-aza. In direct response to this query, we performed a series of experiments in which PPM1D-WT and -mutant cells were treated with two demethylating agents, decitabine, and azacytidine (new **Supplementary Fig. 6d**). Interestingly, our results indicate that we cannot rescue NAPRT expression after treatment with either agent, and perhaps suggest a unique mechanism, involving the demethylation process for the induction of CIMP in *PPM1D* mutant cells. Work is ongoing in our laboratory to further understand the mechanistic basis for these findings.

4. *“Mechanistically, the authors show that PPM1D mutation induces silencing of the NAPRT gene by regulating DNA methylation at its promoter. Since TET mutation is also correlated with a wide range of different cancers, is there mechanistic correlation between PPM1D mutation with TET activity in regulating DNA methylation at the promoter of NAPRT? What is the potential mechanism driving the DNA methylation by mutated PPM1D?”*

The reviewer makes an important suggestion to consider TET proteins in the mechanism of mutant PPM1D-induced CIMP, since these proteins are inactivated in other cancers, and are thought to be a main mechanism for G-CIMP in IDH1/2-mutant gliomas [1]. In the initial version of the manuscript, we assessed methylation and hydroxymethylation levels (often used as a readout of TET activity) using hME-DIP, specifically at the *NAPRT* promoter. These experiments were notable for the detection of increased methylated, but not hydroxymethylated DNA, at the *NAPRT* promoter (see revised **Fig. 3c**). In response to this suggestion by Reviewer 2, we have performed additional studies to assess the extent to which TET proteins play a role in mutant PPM1D-induced CIMP and NAPRT silencing. An examination of global 5-hydroxymethylation revealed no significant differences across matched WT and mutant PPM1D cell lines (new **Supplementary Fig. 6c**), which further argues against a TET-dependent mechanism. In addition, we measured TET protein and mRNA levels, which suggested no mutant PPM1D-dependent differences (data not shown). Thus, we hypothesize that the mutant PPM1D-induced CIMP occurs via a novel, TET-independent mechanism. Work is ongoing in our laboratory to understand the mechanism of action involved in this novel interaction.

5. *“Can treating the mice with NAD⁺ precursors such as NR or NA alter the in vivo sensitivity to FK866 in PPM1D mutant xenograft tumors?”*

The reviewer asks an important and insightful question regarding whether treatment with NAD precursors affects mutant PPM1D-dependent NAMPT inhibitor sensitivity *in vivo*. In the initial version of the manuscript, we demonstrated that NR can rescue NAMPT inhibitor sensitivity *in vitro*. We also tested additional NAD precursors, such as NA and NAM, as requested by Reviewer 2 (see point (2)

above). These experiments revealed that NAM, but not NA, could rescue NAMPTi sensitivity, which is consistent with a model suggesting that loss of NAPRT expression is driving the observed phenotype. It is well established that exogenous NAM antagonizes the effects of NAMPT inhibitors, by providing greater levels of NAMPT substrate, and also by direct competition with these inhibitors ([6]; schematic for these pathways shown in revised **Fig. 2a**). Exogenous addition of NR antagonizes NAMPT inhibitor effects [7], but by a different mechanism: NR enters the NAM salvage pathway via its conversion into the normal NAMPT product, NMN, by NRK, a pathway discovered by co-author Brenner and colleagues [8]. NA also rescues NAMPT inhibitor toxicity via its own unique pathway: conversion into NAMN by NAPRT via the Preiss-Handler salvage pathway [6]. As such, NAPRT levels have been shown to be critical for the ability of exogenously added NA to rescue tumor cells [9]. It thus has been argued that co-administration of NAMPT inhibitors with NA supplementation, specifically in tumors with low or absent NAPRT expression, could be a novel approach to enhancing the therapeutic index between tumor and normal tissue. While the latter finding has been well-established *in vitro*, there is conflicting evidence regarding the feasibility and efficacy of this approach *in vivo* [10]. The presumed mechanistic basis for why this approach may not be effective, is that NA can be processed into metabolites by the liver, which then enter tumor cells and produce NAD, in a NAPRT-independent manner.

The major findings in our manuscript are that mutant PPM1D drives CIMP and NAPRT silencing, which leads to baseline lower levels of NAD, thus rendering the cells sensitive to NAMPT inhibitors. Based on these data, we propose a new biomarker that can select tumors for treatment with this class of agents. As discussed above, it is well known that NAM, NA, and NR can rescue NAMPT inhibitor sensitivity (using both *in vitro* and *in vivo* models), and there is ongoing controversy regarding the role of NA supplementation in NAPRT-deficient tumors. As such, we feel that a series of *in vivo* NAD precursor supplementation experiments would not significantly enhance nor further validate the work that is presented here, and that it would be redundant to previous work demonstrating that NAD precursors can be utilized to antagonize NAMPT inhibitor activity *in vivo*. Nonetheless, we recognize the insightful comments made by Reviewer 2 here, and we appreciate his/her careful review and important suggestions. Work is ongoing in our laboratory to understand the most effective approaches to further widen the therapeutic index between tumor and normal tissue, beyond the use of mutant PPM1D as a novel therapeutic response biomarker as described here.

6. *“The authors show similar results with HCT116 cell line xenografts in vivo. To draw a conclusion that it is a common mechanism in PPM1D mutation tumors, the authors should also provide the evidence showing the mutant PPM1D-induced NAPRT deficiency in HCT116 cells or other cell line harboring PPM1D mutation in vitro.”*

Here, both reviewers 1 and 2, comment on the applicability of HCT116 cells to accurately model mutant PPM1D-dependent NAMPT inhibitor sensitivity. As such, we have decided to remove HCT116 data from the revised manuscript, and instead include data from U2OS and MCF7 cells, in which we demonstrated mutant/aberrant PPM1D-induced NAPRT silencing and NAMPT inhibitor sensitivity *in vitro* (see response to Reviewer 1, point 2; and see new **Supplemental Fig. 5**). In addition, in response to Reviewer 2's request for additional *in vivo* data with one such cell line, we have confirmed NAMPT inhibitor sensitivity in U2OS cells grown as flank tumors in athymic nude mice (see **Supplementary Fig. 8a, b**). Collectively, these data further support the potential clinical relevance of our discovery, as well as its applicability to other solid tumors with PPM1D mutations beyond DIPG and glioma.

References

1. Turcan, S., et al., *IDH1 mutation is sufficient to establish the glioma hypermethylator phenotype*. Nature, 2012. **483**(7390): p. 479-83.
2. Craig, C., et al., *Effects of adenovirus-mediated p16INK4A expression on cell cycle arrest are determined by endogenous p16 and Rb status in human cancer cells*. Oncogene, 1998. **16**(2): p. 265-72.
3. Zhang, L., et al., *Exome sequencing identifies somatic gain-of-function PPM1D mutations in brainstem gliomas*. Nat Genet, 2014. **46**(7): p. 726-30.
4. Wu, G., et al., *The genomic landscape of diffuse intrinsic pontine glioma and pediatric non-brainstem high-grade glioma*. Nat Genet, 2014. **46**(5): p. 444-450.
5. Taylor, K.R., et al., *Recurrent activating ACVR1 mutations in diffuse intrinsic pontine glioma*. Nat Genet, 2014. **46**(5): p. 457-61.
6. Watson, M., et al., *The small molecule GMX1778 is a potent inhibitor of NAD⁺ biosynthesis: strategy for enhanced therapy in nicotinic acid phosphoribosyltransferase 1-deficient tumors*. Mol Cell Biol, 2009. **29**(21): p. 5872-88.
7. Nikiforov, A., et al., *Pathways and subcellular compartmentation of NAD biosynthesis in human cells: from entry of extracellular precursors to mitochondrial NAD generation*. J Biol Chem, 2011. **286**(24): p. 21767-78.
8. Bieganowski, P. and C. Brenner, *Discoveries of nicotinamide riboside as a nutrient and conserved NRK genes establish a Preiss-Handler independent route to NAD⁺ in fungi and humans*. Cell, 2004. **117**(4): p. 495-502.
9. Olesen, U.H., et al., *A preclinical study on the rescue of normal tissue by nicotinic acid in high-dose treatment with APO866, a specific nicotinamide phosphoribosyltransferase inhibitor*. Mol Cancer Ther, 2010. **9**(6): p. 1609-17.
10. O'Brien, T., et al., *Supplementation of nicotinic acid with NAMPT inhibitors results in loss of in vivo efficacy in NAPRT1-deficient tumor models*. Neoplasia, 2013. **15**(12): p. 1314-29.

Reviewers' Comments:

Reviewer #1:

Remarks to the Author:

To improve the scientific rigor of the initial manuscript, the authors have performed additional experiments and re-analyzed the data, which have addressed some of the reviewers' concerns. However, some important concerns have not been addressed:

First, the authors have included several relevant cell lines to demonstrate PPM1D-mutant cells with a "CIMP" phenotype. As described in the manuscript, the authors selected 390 most significant variable probes (SVPs) from 850K methylation array for the cluster analysis and found that 74% probes were hypermethylated in PPM1D-mutant cells and only 24% probes were found hypermethylated in PPM1D-WT cells.

The authors did the measurement of TET enzymatic activity and the sensitivity of DNA demethylating agents between PPM1D mutant and WT cell lines. Unfortunately, both of these two experiments demonstrated negative results and did not show significant differences between PPM1D mutant and WT cell lines. As inhibition of TET and response to DNA demethylating agents are characteristics of CIMP, these negative results cannot support the authors' "CIMP" hypothesis.

In discussion, the authors mentioned the Nature Genetics paper (Zhang 2014) that "As only five PPM1D mutant tumors were present in this diverse cohort (5 of 45 total samples; 11%), the Whole Genome Methylation profiling and analysis performed in this study was likely not sufficient to specifically identify differences in DNA methylation patterns or CIMP in these PPM1D mutant gliomas". The argument is not quite convincing. As it is known that every single IDH mutated tumor is manifested by a G-CIMP phenotype. It is likely that PPM1D is not sufficient to cause G-CIMP.

It is not clear if the algorithm used here is biased. In the work by Turcan et al for IDH1/2-mutant gliomas, they found 30,988 sites were hypermethylated among the 44,334 differentially methylated CpG sites in IDH-mutant cells from 450K array. In the method part, the authors mentioned that "Global CIMP assessments were made using Limma R package of t-test model, with false discovery correction (FDR) and an absolute β -values threshold, to identify probes that reached significance in methylation differential between PPM1D mutant and wild samples (also known as significantly variable probes, or SVPs)". The Nature G-CIMP paper (Turcan 2012) used "unsupervised consensus clustering of the β values was performed with K-means clustering ($K_{max} = 5$) with Euclidean distance and average linkage over 1,000 resampling iterations with random restart on the top 2% of the most variant probes (9,750 probes) using Gene Pattern v.2.0". The Nature paper used unsupervised consensus clustering on the top 2% most variant probes to demonstrate different methylation phenotypes (including CIMP+ and CIMP-) and their relationship with certain genomic variations. In this ms, the authors used probes that reached significance in methylation differential between PPM1D mutant and wild samples. In line 185, the authors mentioned that "Of the 390 most significant variable probes (SVPs), 287 (74%) were hypermethylated in PPM1D mutant lines (PPM1Dtrnc., PPM1DOE, SU-DIPG-XXXV, HSJD-DIPG-007, HSJD-DIPG-008, and HSJD-DIPG-14b), compared to only 107 (24%) hypermethylated in WT cell lines. These data are consistent with a CpG island methylator phenotype (CIMP), being induced by mutant PPM1D". However, it is not clear to what extent the data were consistent with data from the previously published G-CIMP paper. In Nature paper (Turcan 2012), G-CIMP is defined as a phenotype which is "associated with extensive, coordinated hypermethylation at specific loci". Does the methylation phenotype in this study have the similar overall hypermethylation loci? Are these hypermethylation loci the same as those in previous study?

Therefore, it is suggested that the authors apply the same method to analyze the data to determine whether the claim of a mutant PPM1D induced "G-CIMP" is valid. The raw data of 850K methylation array needs to be included as supplementary.

The authors confirmed that the CpG island promoter region of NAPRT were heavily methylated in two of three additional PPM1D-mutant DIPG cell lines. qPCR and Western blotting results are needed to show the loss of PAPRT expression.

The efficacy of NAMPT inhibitor in the three additional PPM1D-mutant DIPG cell lines can enhance the experimental rigor.

The authors cannot rescue the NAPRT expression by treating the cells with DNA demethylating agents. It would be interesting to show that the expression of NAPRT can be rescued by PPM1D inhibition (e.g. GSK2830371 treatment or PPM1D genetic knockout)?

The detailed information about the samples and gene expression raw data used for Fig. 4c and supplemental Fig.8d in the supplementary data need to be provided.

Reviewer #2:

Remarks to the Author:

The authors provided a detailed rebuttal to my criticisms, including the addition of new experiments. Their thorough revision of the paper has improved it considerably. Overall, this is a nice paper that makes some contributions to the field.

As reviewer 2 has accepted our revised manuscript, below is a point-by-point response to the second critique by Reviewer 1:

#1. *“The authors did the measurement of TET enzymatic activity and the sensitivity of DNA demethylating agents between PPM1D mutant and WT cell lines. Unfortunately, both of these two experiments demonstrated negative results and did not show significant differences between PPM1D mutant and WT cell lines. As inhibition of TET and response to DNA demethylating agents are characteristics of CIMP, these negative results cannot support the authors’ ‘CIMP’ hypothesis.”*

Reviewer 1 makes an important point that CIMP is driven by inhibition of TET activity in many cancers that harbor this phenotype, and that these hypermethylation events can typically be reversed by treatment with DNA demethylating agents. While the definition of CIMP has been variable in the epigenetics field (see refs 1-3), we acknowledge his/her point that our finding of consistent, reproducible, and extensive CpG-island hypermethylation in PPM1D-mutant cells likely does not meet the “classical definition” of CIMP. As stated in the rebuttal to our first revision, the findings that we could not detect mutant PPM1D-induced aberrations in TET activity, and that NAPRT expression could not be rescued by treatment with decitabine or 5-azacitidine (which were experiments requested by reviewer 2), suggest a novel mechanism by which PPM1D mutations drive CIMP (revised **Supplementary Fig. 7d, e**). However, in response to Reviewer 1’s valid points, coupled with further analysis of the data here and below, we agree that these findings also further argue against a classical CIMP phenotype. In direct response to the reviewer, we have now defined our phenotype as one consisting of “increased CpG island hypermethylation events” in the revised manuscript.

#2. *“In discussion, the authors mentioned the Nature Genetics paper (Zhang 2014) that “As only five PPM1D mutant tumors were present in this diverse cohort (5 of 45 total samples; 11%), the Whole Genome Methylation profiling and analysis performed in this study was likely not sufficient to specifically identify differences in DNA methylation patterns or CIMP in these PPM1D mutant gliomas”. The argument is not quite convincing. As it is known that every single IDH1 mutated tumor is manifested by a G-CIMP phenotype. It is likely that PPM1D is not sufficient to cause G-CIMP.”*

Here, Reviewer 1 addresses a comment made in the discussion section of our manuscript, which compares our findings to that of a previously published paper (Zhang *et al.* 2014; ref. 4). In this paper from 2014, the authors examined global DNA methylation in a large panel of brainstem gliomas, of which 5 of 45 total samples (11%) contain endogenous PPM1D mutations. While the authors did describe CIMP in the IDH1-mutant gliomas present, they failed to detect a comparable phenotype in the PPM1D-mutant samples. Our discussion highlights the fact that this paper used a number of IDH1-mutant samples which may have over-powered the small number of PPM1D-mutant tumors during analysis. We suggest that this previous paper also makes an assumption that the presence of a PPM1D-induced CIMP would exactly mirror that found in IDH1 mutant gliomas, and would thus be discernable from the analyses performed.

As noted above, we have now defined our phenotype as one consisting of “increased CpG island hypermethylation events”, and thus our revised manuscript no longer proposes that PPM1D mutations are sufficient to cause G-CIMP. However, we maintain that the findings reported in our newly revised manuscript provide a valid and more thorough examination into the effects of PPM1D mutations on CpG

island hypermethylation, beyond what other studies have demonstrated previously. In response to these important comments from Reviewer 1, we have eliminated the direct comparison to G-CIMP as defined by *IDH1*-mutant gliomas in the manuscript, and instead, highlight the differences in hypermethylated probes between *PPM1D*- and *IDH1*-mutant astrocyte cell lines (**Revised Supplementary Fig. 7a-e**). Taken together, we believe that these changes directly address the concerns raised by Reviewer 1 here.

#3. *"It is not clear if the algorithm used here is biased. In the work by Turcan et al for IDH1/2-mutant gliomas, they found 30,988 sites were hypermethylated among the 44,334 differentially methylated CpG sites in IDH-mutant cells from 450K array. In the method part, the authors mentioned that "Global CIMP assessments were made using Limma R package of t-test model, with false discovery correction (FDR) and an absolute β -values threshold, to identify probes that reached significance in methylation differential between PPM1D mutant and wild samples (also known as significantly variable probes, or SVPs)". The Nature G-CIMP paper (Turcan 2012) used "unsupervised consensus clustering of the β values was performed with K-means clustering ($K_{max} = 5$) with Euclidean distance and average linkage over 1,000 resampling iterations with random restart on the top 2% of the most variant probes (9,750 probes) using Gene Pattern v.2.0". The Nature paper used unsupervised consensus clustering on the top 2% most variant probes to demonstrate different methylation phenotypes (including CIMP+ and CIMP-) and their relationship with certain genomic variations. In this ms, the authors used probes that reached significance in methylation differential between PPM1D mutant and wild samples. In line 185, the authors mentioned that "Of the 390 most significant variable probes (SVPs), 287 (74%) were hypermethylated in PPM1D mutant lines (PPM1Dtrnc., PPM1DOE, SU-DIPG-XXXV, HSJD-DIPG-007, HSJD-DIPG-008, and HSJD-DIPG-14b), compared to only 107 (24%) hypermethylated in WT cell lines. These data are consistent with a CpG island methylator phenotype (CIMP), being induced by mutant PPM1D". However, it is not clear to what extent the data were consistent with data from the previously published G-CIMP paper. In Nature paper (Turcan 2012), G-CIMP is defined as a phenotype which is "associated with extensive, coordinated hypermethylation at specific loci". Does the methylation phenotype in this study have the similar overall hypermethylation loci? Are these hypermethylation loci the same as those in previous study? Therefore, it is suggested that the authors apply the same method to analyze the data to determine whether the claim of a mutant PPM1D induced "G-CIMP" is valid."*

Reviewer 1 here astutely recommends the use of a "Top 2% variant probes analysis" to more accurately assess the similarities and differences between our global hypermethylation data, and that from Turcan *et al.* 2012 (as referenced in the manuscript); and he/she questions whether or not our data truly represent G-CIMP. In direct response to these points, we have performed this analysis (revised **Supplementary Fig. 7a, b**), and as described above, we have included a direct comparison between the available data from Turcan *et al.* 2012 and our findings in WT and mutant *PPM1D* astrocytes (revised **Supplementary Fig. 7c**). From this analysis, we demonstrate the predominance of DNA hypermethylation across variable CpG island probesets in *PPM1D*-mutant astrocyte and DIPG models, supporting our previous findings of a mutant *PPM1D*-induced DNA hypermethylation phenotype. Additionally, we highlight the differences in hypermethylated probes between our *PPM1D* mutant astrocytes and those previously published in *IDH1* mutant astrocytes. Overall these new analyses complement our findings presented in Figure 3E, which first demonstrated the presence of global DNA hypermethylation in a diverse panel of *PPM1D*-mutant astrocyte and patient-derived DIPG models, and identify the probes of greatest methylation differences between WT and mutant samples. Tellingly, these studies suggest that mutant *PPM1D*-induced CpG island hypermethylation is distinct from mutant *IDH1/2*-induced CIMP. As addressed in comments **1** and **2** above, we now forgo the direct designation of G-CIMP for *PPM1D*-mutant cells in the revised manuscript. These analyses, coupled with our revised

definition of mutant PPM1D-induced hypermethylation, directly respond to the concerns raised by Reviewer 1 here.

#4. *“The raw data of 850K methylation array needs to be included as supplementary.”*

As the data files for 850K methylation array are extremely large, we believe it is most ideal to deposit this data in a large online database, such as the Gene Expression Omnibus (GEO). We will work towards uploading these files upon final approval of the manuscript.

#5. *“The authors confirmed that the CpG island promoter region of NAPRT were heavily methylated in two of three additional PPM1D-mutant DIPG cell lines. qPCR and Western blotting results are needed to show the loss of NAPRT expression.”*

During the first critique of this manuscript, both reviewers suggested that we test additional cell line models for the presence of mutant PPM1D-induced NAPRT silencing and NAMTPI sensitivity. In direct response to these concerns, we acquired several additional established and patient-derived PPM1D-mutant cell lines, and tested them for this phenotype. As presented in the revised manuscript, we were able to demonstrate clear links between PPM1D mutations and NAPRT silencing and/or NAMPT inhibitor sensitivity in these models, which include: (a) immortalized astrocytes with CRISPR/Cas-engineered heterozygous PPM1D truncating mutations, (b) immortalized astrocytes stably overexpressing WT or mutant PPM1D ORFs, (c) 5 unique primary, patient-derived PPM1D mutant glioma models, and (d) two cancer cell lines with endogenous *PPM1D* aberrations (in total, 9 unique model cell lines; not including several isogenic and matched cell lines with WT PPM1D). In these studies, we demonstrated a direct, robust correlation between *NAPRT* promoter hypermethylation (either by bisulfite promoter sequencing or analysis of whole genome methylation data), decreased *NAPRT* mRNA levels, and loss of NAPRT protein expression, in the majority of these cell lines. In particular, an analysis of the CpG-islands at the *NAPRT* promoter, based on whole genome methylation data and/or bisulfite promoter sequencing, indicate dense *NAPRT* promoter hypermethylation in SU-DIPG-XXXV, HSJD-DIPG-007, PPM1D^{trunc}, PPM1D^{OE}, and HSJD-DIPG-008 cells (**Figs. 3D and 3F**). In parallel, we also demonstrated *NAPRT* promoter hypermethylation in two additional cell lines with PPM1D mutations/aberrations, MCF7 and U2OS (Supplementary **Fig. 5d**). We then confirmed that the observed dense promoter hypermethylation correlates directly with reduced *NAPRT* mRNA and/or protein expression in SU-DIPG-XXXV, PPM1D^{trunc}, PPM1D^{OE}, MCF7 and U2OS cells.

Reviewer 1 requests that we also confirm that the nearly identical pattern of dense *NAPRT* promoter hypermethylation seen in HSJD cell lines, compare to the cell lines noted above, also results in reduced NAPRT expression. To directly address this concern by the reviewer, we now include RNA expression data from these cell lines, presented as z-scores, tested in comparison with a large panel of DIPG cell lines (revised **Supplementary Fig. 6b**). These data confirm reduced *NAPRT* mRNA levels in the PPM1D-mutant models. Unfortunately, direct assessment of protein levels, via western blot, was not able to be completed due to the unavailability of sufficient quantities of protein lysate to perform the analysis.

In summary, we now demonstrate a direct correlation between mutant/aberrant PPM1D and loss of NAPRT expression in the following nine unique cell line models: SU-DIPG-XXXV, HSJD-DIPG-007, HSJD-DIPG-008, PPM1D^{trunc} (two independent single cell CRISPR clones), PPM1D^{OE} (two independent single cell clones expressing WT full-length vs. trunc. ORFs), MCF7, and U2OS cells. These cell line models were acquired from a diverse range of independent laboratories (the Monje, Carcabaso, and Bindra Labs, as

well as commercial sources) across the world. We believe these new data directly address Reviewer 1's concerns here regarding links between mutant/aberrant PPM1D expression and NAPRT silencing.

#6. *"The efficacy of NAMPT inhibitor in the three additional PPM1D-mutant DIPG cell lines can enhance the experimental rigor."*

Reviewer 1 here recommends including viability data from the *PPM1D* mutant HSJD DIPG cell lines upon treatment with NAMPT inhibitors. As mentioned above, due to insufficient growth and proliferation rates of these cell lines, we were only able to achieve the required number of cells for analysis of HSJD-DIPG 007. This data, now presented in revised **Supplementary Fig. 6c**, demonstrate an extreme sensitivity of HSJD-DIPG 007 to treatment of FK866. This strongly supports our previous work and the extension of our findings in immortalized astrocytes to other *PPM1D*-mutant DIPG models. In total, we now demonstrate a direct correlation between mutant/aberrant PPM1D, NAPRT silencing, and NAMPTi sensitivity in 9 unique cell line models: SU-DIPG-XXXV, HSJD-DIPG-007, PPM1D^{trunc} (three independent single cell CRISPR clones), PPM1D^{OE} (two independent single cell clones expressing WT full-length vs. trunc. ORFs), MCF7, and U2OS cells. Collectively, we believe these that these data address the valid concerns regarding experimental rigor raised by Reviewer 1.

#7. *"The authors cannot rescue the NAPRT expression by treating the cells with DNA demethylating agents. It would be interesting to show that the expression of NAPRT can be rescued by PPM1D inhibition (e.g. GSK2830371 treatment or PPM1D genetic knockout)?"*

Reviewer 1 suggests a very novel and interesting experiment, which is designed to further probe the mechanism of action and reversibility of mutant PPM1D-induced *NAPRT* silencing. However, this experiment likely will be subject to multiple confounders: (a) the long-term treatment of *PPM1D* mutant (or amplified) cells with PPM1D inhibitors, such as GSK2830371, is well known to be cytotoxic (refs 5-7), and thus the recommendation to use this drug to rescue *NAPRT* expression likely is not feasible in our *PPM1D*-mutant models; (b) acute genetic knockout of *PPM1D* is also highly cytotoxic to cells, and can result in abnormal levels of DNA damage; and (c) epigenetic marks are often inherited for many cell passages, and can result from "hit and run" based mechanisms (refs 8-10).

The fundamental discovery in our manuscript is that PPM1D/WIP1 mutations, which were first described in the 1980s, silence *NAPRT* expression and confer exquisite NAMPTi sensitivity. These findings are clinically actionable, as they suggest a new biomarker for NAMPTi's, which have been tested extensively in the clinic. Mechanistically, we have found that PPM1D mutations induce CpG island hypermethylation events-leading to *NAPRT* silencing, and ongoing work in our laboratory is focused on studying the exact mechanism of action by which this occurs. As such, we do not feel that additional experiments regarding the reversibility of *NAPRT* silencing would add sufficient mechanistic insights to the work that is presented here. Nonetheless, we thank the reviewer for this insightful suggestion, and we will consider this approach in future studies as we probe deeper in the mechanism(s) of action underlying our novel discovery.

#8. *"The detailed information about the samples and gene expression raw data used for Fig. 4c and supplemental Fig.8d in the supplementary data need to be provided. "*

We agree with this suggestion by Reviewer 1. These data are provided in the publication now cited in the manuscript, and also here (ref 11)

References

1. Toyota, Minoru, et al. "CpG island methylator phenotype in colorectal cancer." *Proceedings of the National Academy of Sciences* 96.15 (1999): 8681-8686.
2. Teodoridis, Jens M., Catriona Hardie, and Robert Brown. "CpG island methylator phenotype (CIMP) in cancer: causes and implications." *Cancer letters* 268.2 (2008): 177-186.
3. Miller, Brendan, Francisco Sánchez-Vega, and Laura Elnitski. "The emergence of pan-cancer CIMP and its elusive interpretation." *Biomolecules* 6.4 (2016): 45.
4. Turcan, Sevin, et al. "IDH1 mutation is sufficient to establish the glioma hypermethylator phenotype." *Nature* 483.7390 (2012): 479.
5. Rayter, S., et al. "A chemical inhibitor of PPM1D that selectively kills cells overexpressing PPM1D." *Oncogene* 27.8 (2008): 1036.
6. Yagi, Hiroaki, et al. "A small molecule inhibitor of p53-inducible protein phosphatase PPM1D." *Bioorganic & medicinal chemistry letters* 22.1 (2012): 729-732.
7. Pechackova, Sona, et al. "Inhibition of WIP1 phosphatase sensitizes breast cancer cells to genotoxic stress and to MDM2 antagonist nutlin-3." *Oncotarget* 7.12 (2016): 14458.
8. Saunderson, Emily A., et al. "Hit-and-run epigenetic editing prevents senescence entry in primary breast cells from healthy donors." *Nature communications* 8.1 (2017): 1450.
9. Amabile, Angelo, et al. "Inheritable silencing of endogenous genes by hit-and-run targeted epigenetic editing." *Cell* 167.1 (2016): 219-232.
10. Niller, Hans Helmut, Hans Wolf, and Janos Minarovits. "Viral hit and run-oncogenesis: genetic and epigenetic scenarios." *Cancer letters* 305.2 (2011): 200-217.
11. Mueller, S, et al. A pilot precision medicine trial for children with diffuse intrinsic pontine glioma—PNOC003: a report from the Pacific Pediatric Neuro-Oncology Consortium. *International journal of cancer* (2019).

Reviewers' Comments:

Reviewer #1:

None